# Modelling six sustainable development transformations in Australia and their accelerators, impediments, enablers, and interlinkages

Cameron Allen ●[1,2] ✉, Annabel Biddulph[1], Thomas Wiedmann ●[1], Matteo Pedercini[3] & Shirin Malekpour[2]

There is an urgent need to accelerate progress on the Sustainable Development Goals (SDGs) and recent research has identified six critical transformations. It is important to demonstrate how these transformations could be practically accelerated in a national context and what their combined effects would be. Here we bridge national systems modelling with transformation storylines to provide an analysis of a Six Transformations Pathway for Australia. We explore important policies to accelerate progress, synergies and trade-offs, and conditions that determine policy success. We find that implementing policy packages to accelerate each transformation would boost performance on the SDGs by 2030 (+23% above the baseline). Policymakers can maximize transformation synergies through investments in energy decarbonization, resilience, social protection, and sustainable food systems, while managing trade-offs for income and employment. To overcome resistance to transformations, ambitious policy action will need to be underpinned by technological, social, and political enabling conditions.

Since 2015, progress towards the global Sustainable Development Goals (SDGs) has been slow and uneven, and some early advancements have been all but derailed by COVID-19[1–3]. As we cross the mid-point of the *2030 Agenda*, the urgent need to accelerate transformations to meet the SDGs is widely acknowledged[4–7]. A key theme for the second global SDG Summit in 2023 was political guidance for transformative and accelerated actions leading up to 2030. Yet in the post-COVID recovery context, government budgets are strained, and financial, human, and technical resources need to be strategically harnessed. While the SDGs provide a blueprint for countries to build back better[3,8,9], deeper transformations will be needed over the longer term to achieve sustainable wellbeing[10]. The field is still exploring the nature of the necessary transformations, how they can be practically accelerated, as well as their potential

beneficial effects and unintended consequences for the SDGs and net zero objectives[11–13].

An important source of evidence that could inform national implementation of the SDGs lies in the recent national scenario modelling literature. Recent national SDGs modelling has nested national scenarios within the global Shared Socio-economic Pathways (SSPs) to analyse specific policy measures, investment needs and implications for a broad suite of SDGs targets[14,15]. However, both the global and national scenario research find that even under a more optimistic sustainability-oriented pathway (e.g. SSP1) progress would still fall well short of goal achievement by 2030 without ambitious and transformative action[15–17].

The development of more transformative national policy scenarios can be informed by recent science-policy research that has

[1]Sustainability Assessment Program, School of Civil and Environmental Engineering, UNSW Sydney, Sydney, NSW 2052, Australia. [2]Monash Sustainable Development Institute, Monash University, Melbourne, VIC, Australia. [3]Millennium Institute, Washington DC, USA. ✉e-mail: cameron.allen@monash.edu

articulated six entry points[18] or six transformations[19] needed to achieve the SDGs. The six entry points feature in the 2023 UN Global Sustainable Development Report[20] which provides science-based advice for national governments and stakeholders to accelerate progress on the SDGs. This provides an integrated organising framework that helps national governments to simplify the 17 goals and support more coherent policy action[21]. However, modelling of multiple SDGs transformations is nascent[16,22] and multi-system interactions remain poorly understood[23]. A further challenge is that accelerating transformations to the SDGs will require not only ambitious national policy action but also due consideration of the social, political and behavioural impediments that have hampered rapid progress to date[24]. Such factors are not often considered in existing modelling studies which assume that impediments to policy action, technology adoption or behaviour change are somehow overcome[25,26]. Complementing national scenario projections with transition storylines[27,28] and pathways[29] approaches provides a means to systematically identify key impediments and explain the enabling conditions needed for techno-economic interventions to succeed.

Here we explore how the six transformations that have been defined as imperative to SDGs achievement could be materialised in the Australian context and their medium- and long-term implications for sustainable development. We aim to explore four critical areas: (i) specific policy packages that can accelerate each transformation, (ii) the individual and combined impact of these policies on achieving the

SDGs and net zero targets, (iii) the complex interactions between the transformations and the SDGs and their synergies and trade-offs, and (iv) important impediments and conditions that determine the success of policy reforms.

To do so we combine national system dynamics modelling to quantify the effects of the transformation policies with socio-technical analysis and transformation storylines to systematically diagnose policy impediments and identify enabling conditions for the ambitious new policies to succeed. This provides evidence to support national operationalisation of the six transformations which provide the best chance for transformative national action to achieve the SDGs[18,20].

The study design comprises two alternative post-COVID-19 recovery pathways for Australia that diverge from a critical juncture after 2020 (Fig. 1, top panel). A Build Back the Same Pathway (BBS) is used as a baseline and signals a return to the pre-COVID status quo (Table 1). Contrasting this, our Six Transformations Pathway (STP) involves the acceleration of six transformations to achieve the SDGs (Fig. 1, main panel; Table 1). Each transformation (labelled T1 to T6 in Fig. 1) includes a set of quantitative policy levers used in the model projections and a qualitative sociotechnical analysis and transformation storyline which diagnoses key impediments and describes how various actors improve critical enabling conditions for the successful implementation of the policy levers. We use an upgraded national integrated system dynamics model (iSDG-Australia v2.0)[30] to model relationships and project and evaluate progress made by the two

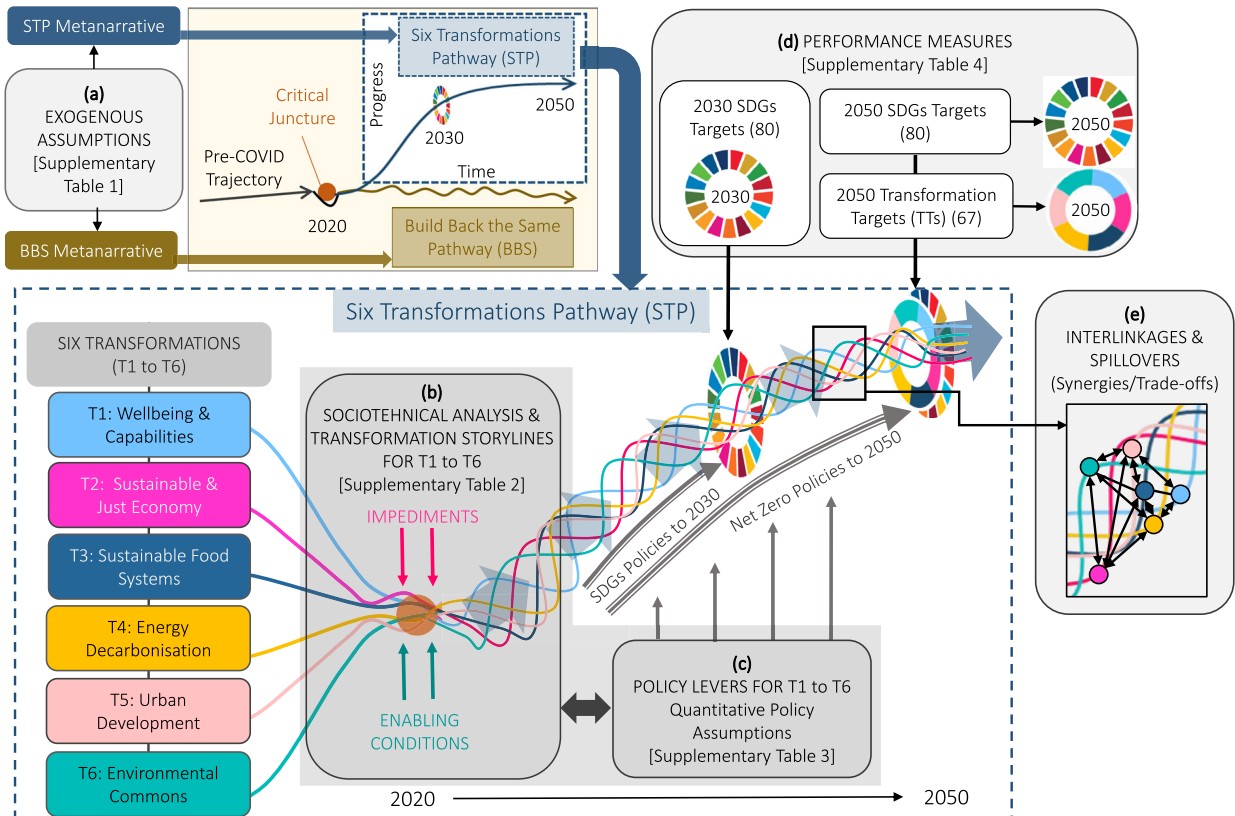

**Fig. 1 | Diverging pathways and the framework for the Six Transformations Pathway (STP).** At the top left of the figure, the baseline Build Back the Same Pathway (BBS) and the STP diverge from a critical juncture after 2020. Each pathway is guided by a metanarrative and has different exogenous assumptions regarding global drivers (**a**). Inset main figure shows the STP combining six transformations (T1 to T6). Each transformation includes a qualitative storyline (**b**) which reviews impediments to transformation, emerging positive seeds of change, and enabling conditions for acceleration. Each transformation also includes a set of quantitative policies (**c**) which are used as inputs to parameterise the model. This

includes a diverse suite of policies to accelerate progress on the SDGs which end in 2030 and net zero policy assumptions which continue until 2050. The iSDG-Australia 2.0 model is used to model relationships and projects each transformation individually as well as the combined STP (and the BBS) through until 2050 and evaluates performance (**d**) on the SDGs targets in 2030 and in 2050 along with a subset of transformation targets (TTs) in 2050. Finally, multi-system interactions (**e**) are also evaluated through the model projections including between the six transformations and the SDGs.

**Table 1 | Post-COVID metanarratives for the two pathways**

| | |
|---|---|
| Build Back the Same (BBS) | Australia's post-COVID pathway is based on a gas-led recovery and return to the status quo which pays little attention to the SDGs and leaves little room for structural change. In this recovery pathway, we assume that political inertia in addressing climate change remains unchecked, opportunities for governance improvements such as a national anti-corruption commission are postponed or watered down, and that government economic stimulus tapers off after 2021. Government expenditure and revenue settings largely return to pre-COVID levels (Supplementary Table 3), with the inclusion of additional legislated tax cuts resulting in a less distributive tax system. This places Australia largely on a business-as-usual recovery pathway which sees a continuation of pre-COVID trends. |
| Six Transformations Pathway (STP) | With the convergence of multiple compounding crises, a pervasive narrative emerges in Australia around the need for structural change and to build back better using the SDGs as a roadmap. This gains support from powerful actors and coalitions which legitimizes stronger policy action and a more interventionist approach by governments in setting targets, launching missions, shaping markets, investing in infrastructure, and regulating business. The crises weaken resistance from incumbent actors and coalitions for change gather momentum. Elections shift the political landscape and disrupt existing power balances. Governments form strong coalitions with powerful actors from business and civil society to prosecute a sustainable recovery based on a six-pronged transformation agenda guided by the SDGs and net zero ambitions. A legislated anti-corruption commission tempers the influence of vested interests and improves government effectiveness and political stability. To accelerate progress, governments make additional targeted investments through to 2030 (Supplementary Table 3) which are packaged around six transformative 'missions' with clear goals and targets. Over the longer-term beyond 2030, ambitious action continues to reach net zero by 2050. |

pathways towards all 17 SDGs in 2030 and 2050 as well as a subset of transformation targets (TTs). Interlinkages and spillover effects between the transformations and the SDGs are also evaluated. To support robustness, we analyse the sensitivity of the projections to changes in exogenous global assumptions. In this way, we include common features of the XLRM framework[31] (policy levers, exogenous factors, model relationships, and performance measures). The approach used also bridges modelling and transition studies[25] to provide a comprehensive analysis of the transformation pathway.

## Results

### Overview of the study approach and alternative pathways
Further details of the study design and model parameterisation are provided in 'Methods' and Supplementary Tables. In brief, the study proceeded in several steps which included some iteration and which were informed by recent advancements in methods for enriching quantitative modelling with empirical findings from socio-technical transitions research[25,27,32]. Firstly, we developed a metanarrative (Table 1) for each pathway including different exogenous assumptions of landscape trends (Supplementary Table 1 and Methods: 'Critical juncture and pathway metanarratives'). Secondly, for each transformation (T) we reviewed empirical research, policies, reports and other literature to inform policy choices and to provide a socio-technical analysis of current regime conditions, impediments, and promising niche innovations. A standardised template was used for each transformation informed by an extensive review of the literature on accelerating SDGs transformations[24] (Supplementary Table 2; see Methods: 'Socio-technical analysis and transformation storylines').

The first two steps informed the development of quantitative policy settings for each pathway which serve as the key inputs to parameterise the model and project the two pathways to 2050 (see Supplementary Table 3 and Methods: 'Elaboration of quantitative policies and transformation storylines'). For BBS, existing tax, subsidy, expenditure, and other settings remain in line with recent trends over the period to 2050. For STP, each transformation includes an ambitious package of SDGs policies from 2021 to 2030, beyond which policy settings return to trend except for net zero policy assumptions which are continued to 2050 (Fig. 1). In a fourth step, the quantitative projections for each transformation were confronted with the socio-technical analysis of key impediments. A future-oriented qualitative storyline was then constructed for each transformation which enriches the modelling by describing how impediments are overcome through particular socio-technical enabling mechanisms (e.g. learning processes, changing coalitions, public support, institutional change) (Supplementary Table 2; see also Table 2 in 'Methods'). The quantitative policy settings were also

revisited to ensure consistency with the storylines, in particular regarding the scale and speed of change.

The performance of the final model projections for BBS and STP (and each individual transformation) was then benchmarked against the 2030 and 2050 targets (see Methods: 'Targets and method for assessing progress'). For the SDGs, the model includes a set of 80 unique SDG indicators covering all 17 goals with target values for 2030 and (more ambitious values) for 2050 (Fig. 1; see Supplementary Table 4). A subset of 67 of these indicators was also used to evaluate progress made on the six transformations by 2050, which we label as transformation targets (TTs) (Fig. 1). Each transformation was allocated a unique set of TTs based on thematic relevance (e.g. energy-related SDG indicators are used as TTs for T4 on energy access and decarbonisation). We avoid duplication of TTs across the different transformations and exclude means of implementation indicators (from SDGs 16 and 17) as they could apply to all transformations (see Methods: 'Targets and method for assessing progress').

While there is considerable overlap between the 2050 SDGs targets and the TTs, they assist the analysis as they are aggregated differently (i.e. at the goal level for the 17 SDGs or at the transformation level for the six Ts). Using this evaluation approach has several advantages. It enables us to compare the performance of the two pathways on all 17 SDGs in 2030 and 2050 and to decompose the comparative impact of each individual transformation on the SDGs (see Methods: 'Methods for assessing interlinkages'). It also enables us to understand the comparative progress made on enabling each transformation, the acceleration dynamics resulting from different packages of policies, and spillover effects from one transformation to another (both positive and negative). This provides insights on the complex interlinkages among the six transformations and the SDGs and important dynamics over time.

### COVID-19 and cascading crises
In Australia, the COVID-19 pandemic arrived on the heels of the 2019 catastrophic bushfires and was succeeded by unprecedented flooding. Together the crises had a profound impact, ending 30 years of economic growth, temporarily halting net migration, disrupting supply chains, and exacerbating weaknesses in Australia's development model[33]. In response to the pandemic, the Australian government implemented emergency economic stimulus of around 15% of gross domestic product (GDP)[34,35]. A large share was provided as social transfers directly to individuals, households, and businesses. New infrastructure investment targeted transport, construction, and the oil and gas sector, with green stimulus making up less than one percent in 2020[36]. The socio-economic effects of COVID-19[37,38] and the emergency stimulus measures are incorporated into our modelling

projections, showing a V-shaped economic recovery, an A-shaped peak in annual unemployment, and a V-shaped dip in total greenhouse gas (GHG) emissions, among other effects (Supplementary Fig. 1a–d). Note that the shape refers to a short-term fall (or increase) followed by a return to trend.

Moving ahead, Australia faces a range of possibilities for its medium-term recovery. At the end of 2020, the national Government's announcement of a gas-led recovery signalled a return to the status quo which forms the basis for our Build Back the Same Pathway (BBS) (Table 1). The long-term outlook for BBS is one of stagnation and slow decline in progress towards sustainable development (Fig. 2a). Following a brief jump in SDG progress in 2020 resulting from the stimulus expenditure, progress subsequently slows with only modest gains on the SDGs by 2030 (59% average progress across all goals and targets). The best performing goals are education (SDG 4), health (SDG 3) and water (SDG 6), while progress lags on environmental goals (SDGs 14 and 15). Australia also performs well on indicators relating to government revenue and debt (SDG17). However, these gains are not sustained and over the longer-term to 2050, progress on the SDGs tapers off and then declines to ~55%, largely due to worsening climate change impacts and a lack of investment in adaptation and resilience (Fig. 2b).

## A critical juncture: pathways diverge

A return to the status quo is not a given. Research shows that crisis events can lead to a critical juncture and provide opportunities for systems transformation[39,40] (Fig. 1). Our Six Transformations Pathway (STP) is premised on such a juncture occurring in Australia from 2021/2, triggered by the compounding crises described above (Table 1). On this pathway, government acts decisively to accelerate six transformations to achieve the SDGs by 2030. The STP incorporates a range of additional policies which are packaged into six individual transformations (T1 to T6) (Fig. 1; Supplementary Table 3). These cover a broad mix of policies, including tax reform, additional spending on social services and transfers, and scaled-up investment in sustainable energy, industry, transport, agriculture, and infrastructure. Annual government expenditure increases by 7.5% on average or approximately AUD87 billion per annum which is offset through additional revenue measures (Supplementary Fig. 1d). This involves a continued but scaled-down stimulus following the immediate response to COVID-19 from 2021 to 2030, beyond which net zero policy assumptions are continued to 2050.

The medium-term outlook for the STP is one of acceleration reaching 82% progress on all SDGs by 2030 (Fig. 2a). Strong performance is evident across all goals, with education (SDG 4), sustainable energy (SDG 7), sustainable cities (SDG 11), climate change (SDG 13), marine biodiversity (SDG 14) and partnerships (SDG 17) achieving over 90% progress. Performance is marginally lower than BBS on per capita GDP and disposable income (SDG8), partly because of higher government revenue (tax) settings. The additional investment to 2030 and continued measures to achieve net zero have long-term positive effects, with performance on the SDGs projected to continue beyond 2030 and rise to reach 89% average progress on all goals and targets by 2050 (Fig. 2b). When we categorise the targets by domain, progress in 2050 is relatively balanced across economic (77%), social (88.1%) and environmental (85.3%) targets (Supplementary Fig. 2). Results from the sensitivity analysis show that these projections are relatively robust to changes in the global outlook, with 95% of (6000) simulations ranging between 73% and 91% average progress in 2050 (Fig. 2b; see also Supplementary Fig. 3a).

## The acceleration dynamics of the six transformations

The additional progress on the SDGs in the STP results from the combined effects of policy settings contained within the six transformations (Supplementary Table 3), with progress accelerating rapidly over the period to 2030. Here we measure the progress made on each of the individual transformations using a unique set of transformation targets (TT) associated with each transformation (which we label TT1 to TT6) (Supplementary Table 4). These targets provide a consistent long-term framework to measure and evaluate progress on each transformation and a means to explore multi-system interactions and spillovers between the transformations (see Methods: 'Targets and method for assessing progress').

Empirical research shows that successful transformations often go through phases of emergence, acceleration and stabilization, taking the shape of an S-curve[5,41–44]. Acceleration occurs when a tipping point is crossed, shifting from incremental to rapid non-linear progress[45]. Acceleration dynamics are evident to varying degrees in the projections for the six transformations in the STP, with the shaded area in the charts (Fig. 3a) highlighting a period of acceleration resulting from the SDGs policies implemented over the period to 2030 as well as from tipping points (e.g. price parity) for critical technologies such as renewables and electric vehicles (Supplementary Fig. 4).

The projections in Fig. 3a are for the average progress made by the STP and BBS towards the 2050 TTs for each transformation. In the STP, progress on each of the six transformations accelerates to varying degrees (Fig. 3a), closing in on 100% achievement of TTs in most cases, but with more limited progress for T2 (economy, 79%) and T3 (food, 87%). For the BBS, projected progress is particularly limited in transforming the economy (T2), food systems (T3) and the environmental commons (T6), and the projections show different pathways of early lock-in, stagnation or decline. Overall, the projections for STP are consistently ahead of BBS across each transformation (Fig. 3b). Total average progress towards all six transformations reaches over 92% for STP, well ahead of BBS which locks-in by 2030 at under 62% (Fig. 3c). These results are robust to changes in the global outlook (sensitivity analysis range of 85% to 93%) (Fig. 3c; see also Supplementary Fig. 3b). Projections to 2050 for the BBS and STP for all 67 indicators used in the transformation targets are available in Supplementary Data 1.

The modelling results are consistent with empirical research on transitions which suggests that acceleration is contingent upon decisive government action[5,7,46,47]. Such coordinated, government-led development is a common scenario archetype[48,49]. However, a frequent criticism[50,51] is that governments are unlikely to act decisively unless the right conditions are in place[7] to overcome impediments such as political feasibility, vested interests, social acceptance, and barriers to technology diffusion[46,52,53]. These issues are addressed in our transformation storylines (Supplementary Table 2) which describe how transformative change could unfold across key systems. We further explore these broader issues here, combining insights from both the modelling projections and transformation storylines for T2 (economy), T3 (food) and T4 (energy) as examples.

## A sustainable and just economy as a central pillar (T2)

Our projections suggest that progress towards a sustainable and just economy (T2) (Fig. 3a) can be accelerated by reforming Australia's social protection and tax systems and incentivizing sustainable production and consumption systems (Supplementary Table 3). These systems are in a state of inertia due to a range of impediments diagnosed in our sociotechnical analysis of the literature (Supplementary Table 2). For example, divisions in Australia's economic debate are sharp with unresolved conflicts between economic growth and wider policy goals, the profits of businesses and the wages of workers, public ownership versus privatisation, and wealthy inner cities versus struggling outer suburbs and regions[33,54]. Attempts at tax reform have met with fierce resistance from businesses, workers, property owners, retirees, and shareholders[55]. Large sunk investments and policy support for incumbent export-oriented industries crowds out green investment and new business models[56,57].

However, the sociotechnical analysis also finds that conditions for systems change may be taking shape in Australia, with positive signs of

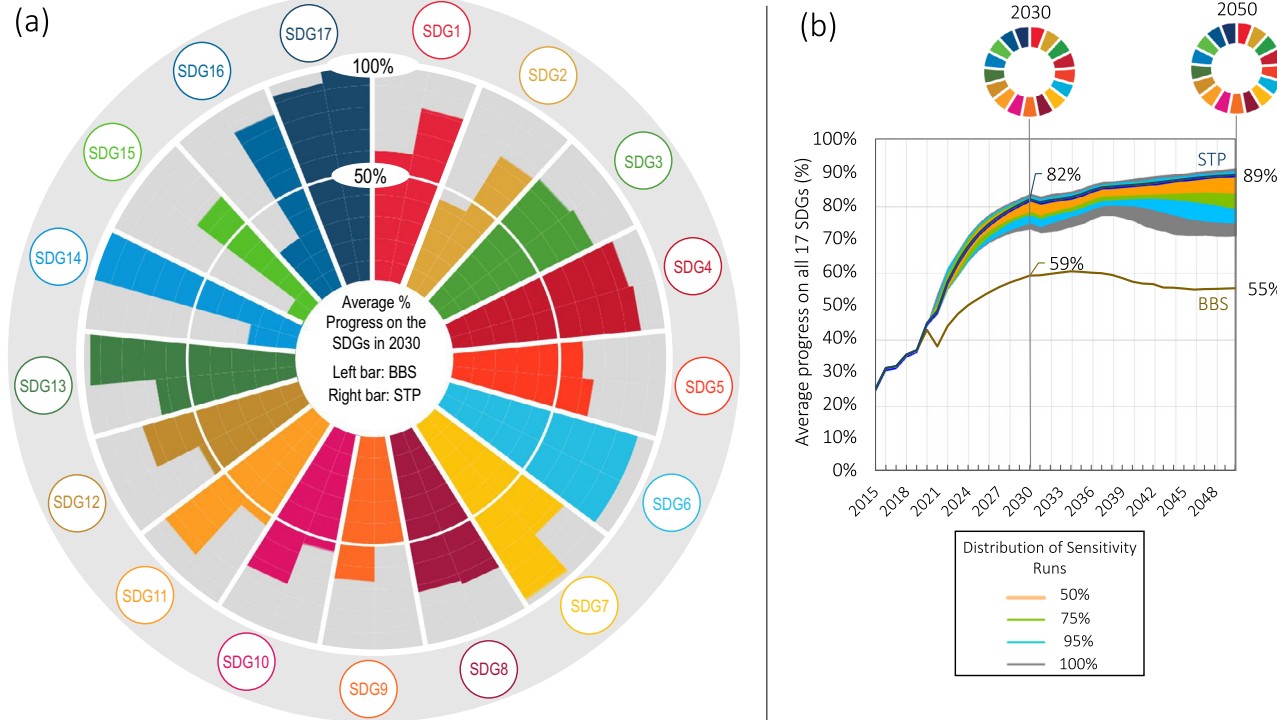

**Fig. 2 | Projected performance of the Build Back the Same Pathway (BBS) and Six Transformations Pathway (STP) on the Sustainable Development Goals (SDGs). a** Average % progress on each of the SDGs in 2030 presents the average progress of the BBS and STP towards each of the 17 SDGs in 2030. Progress is evaluated based on a set of 80 unique indicators distributed across each of the 17 goals and target values for 2030 (Supplementary Table 4). Coloured bars represent the average percentage of progress towards the targets for each goal (0–100%). Left bar = BBS; Right bar = STP. The 17 SDGs are numbered as follows: SDG1 = No Poverty; SDG2 = Zero Hunger; SDG3 = Good Health and Well-being; SDG4 = Quality Education; SDG5 = Gender Equality; SDG6 = Clean Water and Sanitation; SDG7 = Affordable and Clean Energy; SDG8 = Decent Work and Economic Growth; SDG9 = Industry, Innovation and Infrastructure; SDG10 = Reduced Inequalities; SDG11 = Sustainable Cities and Communities; SDG12 = Responsible Consumption and Production; SDG13 = Climate Action; SDG14 = Life Below Water; SDG15 = Life on Land; SDG16 = Peace, Justice and Strong Institutions; SDG17 = Partnerships for the Goals. **b** Aveage % progress on all 17 SDGs in 2030 and 2050 presents the projected average performance of the BBS and STP on all SDGs over the period 2015 to 2030 and 2050 using the full set of 80 indicators and target values. In Panel (**b**), these are overlayed with the results from the sensitivity analysis (Methods: 'Sensitivity analysis') which provide a distribution of sensitivity runs (50%, 75%, 95% and 100%) for 6000 simulations adjusting 10 exogenous and uncertain parameters.

new solutions, coalitions, and political momentum for reforms (Supplementary Table 2). Several proposals for wellbeing budgets are moving forward[54,58], building on experience in New Zealand. Most large businesses are now reporting on the SDGs, Australia's $3.3 trillion superannuation sector is aligning investments to the goals[59,60], and powerful actors are investing in emerging green technologies and mega projects[61,62]. There is growing momentum for much-needed tax reform[55,63,64]. In our T2 storyline (Supplementary Table 2), these green shoots accelerate as actors orient towards shared goals of halving poverty, reducing inequality, tackling debt, and a green economy.

For T2, specific policies parameterised in the model include additional expenditure on subsidies and transfers with more progressive distribution, increased taxes on income/profits, trade and consumption, and investment in green manufacturing, amongst others (see T2 in Supplementary Table 3). By 2050, STP projections for transformation targets associated with T2 (economy) approach an average of 79% progress (Fig. 3a), well above the baseline of 57% under the BBS, however, less than other transformations. As a result of the SDGs policies included in the STP (Supplementary Table 3), poverty rates decline by more than half, income inequality declines by 30% (Gini coefficient from 0.33 to 0.23), green investment boosts manufacturing output by 90%, along with reductions in domestic material consumption (−33%) and material footprint per unit of output (−22%) (Supplementary Fig. 5a–e).

Additional revenue from tax reforms in T2 funds a broad suite of government investments including an increase in social transfers and redistribution to address poverty and inequality (T2), increased expenditure on health, education and resilience (T1), investments to transition to sustainable food systems (T3), energy decarbonization (T4), and sustainable cities (T5), as well as for increased biodiversity protection and reforestation (T6). The tax reforms included in T2 are therefore foundational for all six transformations.

**Strong synergies from a food system transformation (T3)**
In the wake of multiple crises, our storyline for T3 (food) (Supplementary Table 2) focuses on enabling conditions that unlock the transition towards a healthy, regenerative, and equitable food system. Our sociotechnical analysis revealed that impediments to change include the unsustainable growth imperative which drives the need to maximise output at the lowest cost[57,65,66] and which incentivises industrial farming, ultra-processed foods, a concentrated retail sector, and large volumes of packaging and food waste[67,68]. Other impediments identified in the literature include sunk investments which create vested interests that resist change[65,69], modern lifestyles that rely on fast, cheap, and convenient foods[69], and eradication of sustainable Indigenous agricultural practices[69]. Current policy settings provide little support or incentives for farmers to adopt sustainable practices.

Informed by our sociotechnical analysis, the T3 storyline (Supplementary Table 2) describes how systems change comes in response to cascading crises. Enabling conditions include new shared goals, shifting narratives and preferences for healthy diets and lifestyles, new

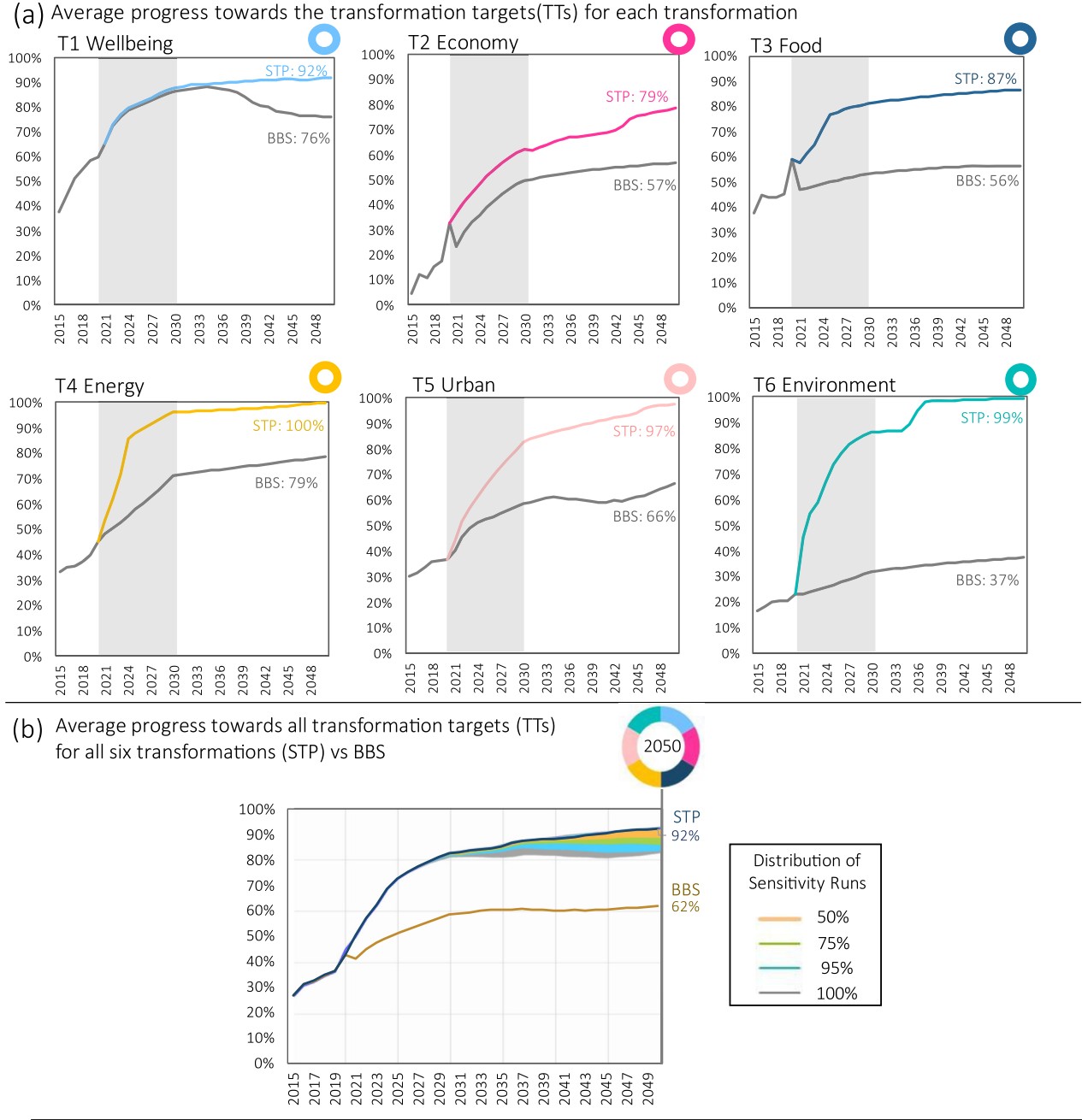

**Fig. 3 | Projections of average performance of the Build Back the Same Pathway (BBS) and Six Transformations Pathway (STP) on the transformation targets.** Panel (**a**) presents the projections for STP and BBS showing the average percentage progress towards the unique set of 2050 transformation targets (TTs) associated with each transformation (T1 to T6). The shaded area shows the acceleration period from 2021–2030 which coincides with the duration of the SDGs policies. Panel (**b**) presents the projection results for the two pathways for the average progress towards all TTs. Note that the TTs comprise a set of 67 indicators which are a subset of the 80 SDGs indicators used to evaluate progress towards the SDGs, as depicted in Fig. 2. They are considered most relevant for measuring progress on each transformation, excluding the means of implementation goals and avoiding duplication of indicators across transformations. Projections are overlaid with the results from the sensitivity analysis (Methods: 'Sensitivity analysis') which provide a distribution of sensitivity runs (50%, 75%, 95% and 100%) for 6000 simulations adjusting 10 exogenous parameters.

business models supporting local supply chains, disruptive emerging technologies, and a growing regenerative agriculture movement that builds into a coordinated coalition for change[65,69–72]. The food system reorients towards sustainable targets, and over the long-term new technologies such as meat alternatives and feed substitutes are scaled up to reduce GHG emissions.

With these conditions in place governments act decisively, providing new incentives, extension services and financing options to support farmers to adopt regenerative practices. Policies for T3 that are implemented in the model (Supplementary Table 3) include an increase in investment in sustainable agriculture and water efficiency to 2030 as well as ambitious reductions in emissions from livestock (−83%) and cropping (−27%) by 2050[72], largely resulting from technological advancements. By 2050, STP projections for eight transformation targets associated with T3 reach an average of 87% progress, well above the baseline of 56% for BBS (Fig. 3a). Important gains are

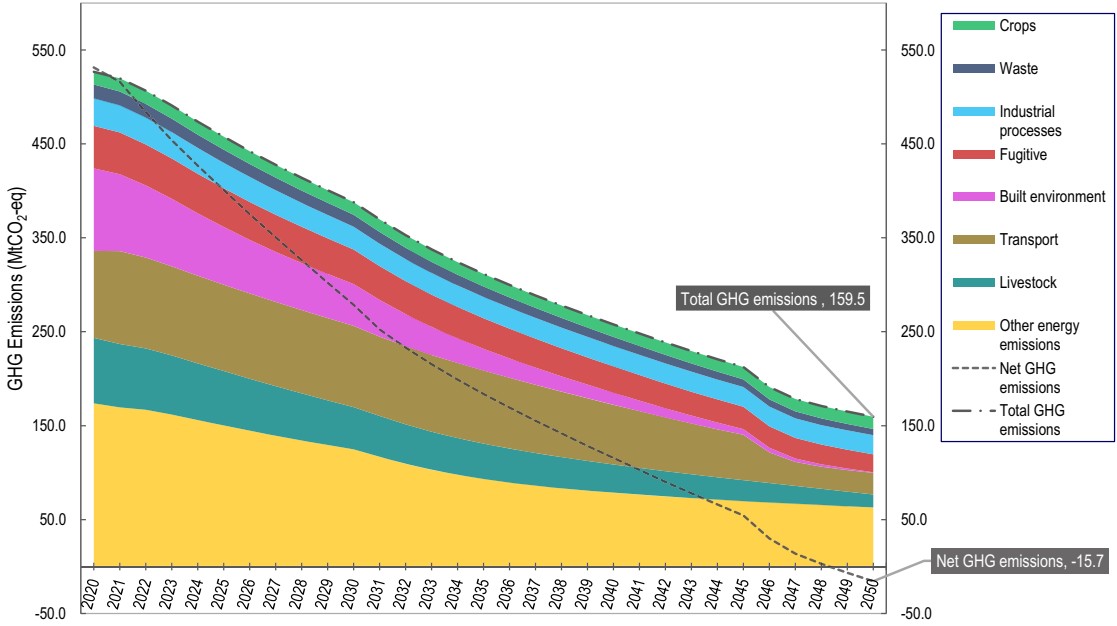

**Fig. 4 | Six Transformations Pathway (STP) projections for total Australian greenhouse gas (GHG) emissions (Mt CO₂-eq) by sector from 2020 to 2050.** Stacked chart presents model projections for GHG emissions from eight main sectors: crops, livestock, waste, industrial processes, fugitive emissions, built environment, transport and other energy emissions. Total GHG emissions and Net GHG emissions are also shown, the latter of which accounts for carbon offsets from reforestation.

made in the share of harvested area sustainably managed along with reductions in fertilizer consumption and population below the food poverty line (Supplementary Fig. 6a–d).

### An accelerating energy transition takes off (T4)

The literature and sociotechnical analysis for T4 (Supplementary Table 2) highlights that Australia's energy transition has begun to accelerate after almost 30 years of inertia, resistance and national policy uncertainty[56,73]. A key impediment has been that Australia's abundant fossil fuel resources support a powerful sector which has succeeded in weakening, delaying or shaping policy responses to climate change[56]. Incumbents have worked to ensure that electricity market rules favour large, centralised fossil fuel generators, making market entry harder for decentralised renewable sources[56,74]. High fossil fuel subsidies contribute to an uneven playing field, and Australian lifestyles are structured around the consumption of readily available, reliable and affordable energy[33].

In our T4 storyline (Supplementary Table 2), the decarbonisation of the energy system takes off, with key drivers including community concerns about climate change, an aging coal generation fleet, and mature and competitive alternative technologies[72,74,75]. The strong coalition of political actors, business, unions and the community gathers momentum. All national and state governments legislate ambitious reduction targets for 2030 and net zero by 2050. Detailed technical roadmaps by research institutions provide clear decarbonisation pathways which governments and stakeholders put into action[72–76,77]. Important triggers for acceleration include price-parity tipping points for renewables and other technologies which provide solutions that governments can push.

Underpinned by this storyline, the policies implemented in the model for T4 include increased investment in small- and large-scale renewables and industry energy efficiency to 2030, and longer-term settings to 2050 for accelerated fuel switching, capture of waste and fugitive emissions, and deliberate phase out fossil fuel generation (Supplementary Table 3). Based on these, the STP projections for T4 reach full achievement of associated TTs by 2050 (Fig. 3a). This includes the share of renewables in electricity and in total final energy consumption reaching 100% and 70%, respectively (Supplementary Fig. 7a, b).

### Multi-system interactions deliver net zero by 2050

The STP projections show how efforts to achieve the SDGs and accelerate the six transformations are also coherent with the longer-term objective of achieving net zero and reducing emissions across all sectors (Fig. 4). The transition of the energy system (T4) paves the way for important gains across other transformations including decarbonising Australia's built environment and transport sectors (T5), enabling green manufacturing (T2), and reducing energy emissions in the agriculture sector (T3). Also of relevance for decarbonisation are the policies included in the model for T5 (urban), which incentivize the electrification of buildings, mandate timber buildings and resource recovery targets, provide electric vehicle subsidies and tax rebates, invest in charging infrastructure, and improve waste management through circular economy initiatives (see T5, Supplementary Table 3). The model projections show that the combined effect of these transformations results in a 72% reduction in total GHG emissions between 2016 and 2050 (Fig. 4). Combining these with large investments in reforestation (see T6, Supplementary Table 3) and climate change adaptation (see T1, Supplementary Table 3) places a net zero outcome within reach by 2050 (Fig. 4). The results from the modelling suggest that accelerating progress on T4 can create a pull effect with many positive spillovers for other transformations.

### Complex synergies and trade-offs among the six transformations

The modelling results reveal that T4 (energy) has the greatest measurable synergistic policy effects on other transformations, enabling a green transition in T2 (economy), T3 (food), T5 (urban) and T6 (environment). The matrix heatmap in Fig. 5a presents the modelled effects of the policy settings in each transformation (T1 to T6, modelled individually) on progress towards the transformation targets (TT) for each transformation (TT1 to TT6; Methods: 'Assessing

**(a)** Additional projected progress towards transformation targets (TT) for each transformation (T1 to T6) compared to BBS (%)

| ROW | | TT1 ◯ | TT2 ◯ | TT3 ◯ | TT4 ◯ | TT5 ◯ | TT6 ◯ | Total Spillovers |
|---|---|---|---|---|---|---|---|---|
| A | T1 | 14.9% | 5.9% | 6.6% | 1.0% | 8.4% | 2.8% | 24.6% |
| B | T2 | 1.9% | 12.1% | 0.7% | -0.1% | 0.7% | 1.1% | 4.3% |
| C | T3 | 0.7% | 2.3% | 15.8% | 0.3% | -1.0% | 14.7% | 17.1% |
| D | T4 | 1.0% | 14.3% | 0.9% | 17.4% | 6.0% | 8.6% | 30.8% |
| E | T5 | 0.1% | 3.7% | 0.9% | 7.4% | 18.7% | 4.6% | 9.2% |
| F | T6 | 0.7% | 2.5% | 0.8% | 0.3% | -0.4% | 40.0% | 4.0% |
| G | STP | 15.7% | 22.1% | 30.3% | 21.0% | 30.9% | 62.1% | |
| H | Sum of spillovers for each TT | 4.4% | 28.7% | 9.9% | 8.9% | 13.7% | 31.7% | |
| I | TT spillovers in the STP | 0.7% | 10.0% | 14.5% | 3.6% | 12.2% | 22.1% | |

**(b)** Additional average projected progress towards all TTs compared to BBS (%)

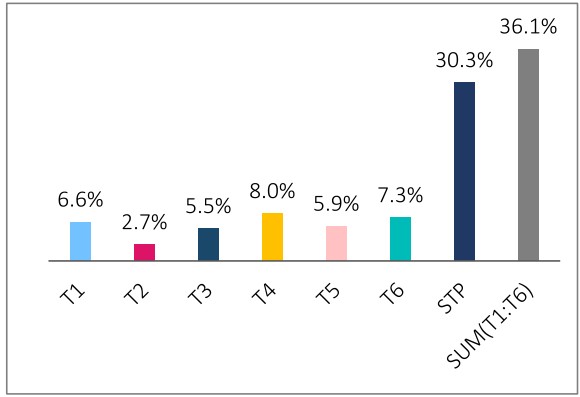

**(c)** Projected transformation interactions – aggregation losses and gains (Row I-H in (a) above) (%)

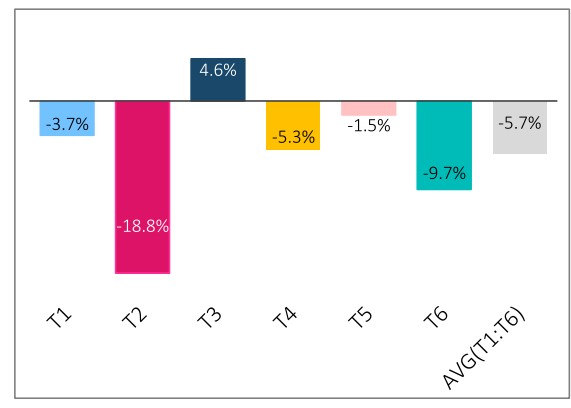

**Fig. 5 | Analysis of interactions between the transformations based on model projections for 2050.** Panel (**a**) presents the simulated effects of the policies and investments in each individual transformation (T1 to T6) on the transformation targets (TT) associated with each transformation (TT1 to TT6) in 2050. Percentages in the coloured cells reflect the average additional (or worsened) progress towards the targets compared against the baseline Build Back the Same Pathway (BBS) simulation. Positive values reflect synergies and negative values reflect trade-offs. The Spillovers column presents the sum of synergies and trade-offs between a single transformation and the other five transformations. The last three rows present calculations for spillover and aggregation effects. Row G presents the additional average progress made on the TTs by the aggregate Six Transformation Pathway (STP) compared to BBS. Row H sums the spillover effects from individual transformations (Ts) on the TTs (e.g. for $TT1 = (SUM(T1:T6)-T1)$. Row I presents the spillovers embedded in the aggregate STP (e.g. for TT1 = STP-T1). Panel (**b**) presents the additional average progress made by each T and STP on all TTs in 2050 compared against the BBS. When these projections for T1 to T6 are summed (final bar = 36.1%) they are greater than STP (30.3%) which gives a difference of -6%. Panel (**c**) visually displays the calculated aggregation losses or gains (Row I-Row H from 5(a)) (Methods: 'Assessing interlinkages'). Where these values are negative they present aggregation losses as the transformations are layered upon one another, and where they are positive they represent multiplier effects unlocked by the multi-system interactions. The final grey bar (AVG(T1:T6)) is the average of these values for all Ts.

interlinkages'). Positive percentage points highlight synergies between transformations, negative values present trade-offs, while values in the spillovers column are net sums from one transformation to all other transformations.

For example, reading from left (rows) to right (columns) (Fig. 5a), the policy effects projected for T1 (wellbeing) also advance progress towards all other transformations as measured by their TTs, particularly TT5 (+8.4 percentage points through more resilient urban systems) and TT3 (+6.6 percentage points through more resilient food systems). The total net benefits of spillovers from T1 on the other five transformations amount to +24.6 percentage points progress (final column). Moving down a row (Fig. 5a), the policies in T2 result in more limited positive spillovers to other transformations (+4.3 percentage points) while T3 projects strong synergies with TT6 on biodiversity (+14.7 percentage points), largely due to improved land management. Overall, T4 on energy decarbonisation has the largest positive spillovers on all other transformations (+30.8 percentage points).

Some trade-offs are also evident in these results (negative values), for example between T3 (food) and T5 (urban). The largely positive values suggest that synergies generally compensate for trade-offs.

However, if we sum the individual gains above the BBS for all transformations (Fig. 5b, +36 percentage points above BBS) and compare them against the aggregate results for the STP (+30 percentage points above BBS or 92–62%), there is a discrepancy of around 6 percentage points progress which is lost when the six transformations are layered upon one another. This can be seen in Fig. 5b which shows the additional progress made by each T and the STP compared to BBS. Summing these for T1 to T6 reaches 36% compared to 30% when combined in STP, or a difference of ~6 percentage points.

This indicates areas of conflict or duplication in the complex multi-system interactions which we unpack further in Fig. 5c. Each bar presents aggregation losses (negative values) or multiplier effects (positive values) which occur when the six transformations are layered upon one another (Methods: 'Assessing interlinkages'). We calculate these by subtracting the sum of spillovers for each transformation from the spillovers in the aggregate STP (or *Row I-H* in the matrix in Fig. 5a). The final grey bar in Fig. 5c again shows the average aggregation losses of ~6 percentage points.

The aggregation losses more clearly identify potential trade-offs associated with each transformation. Trade-offs are particularly large

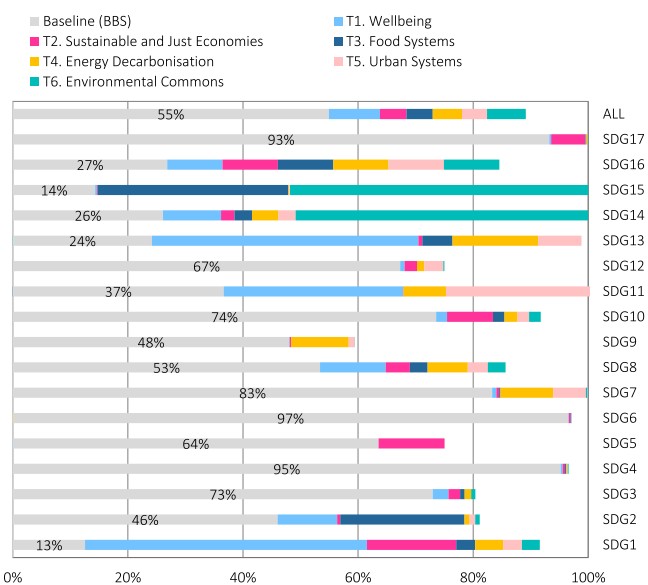

**Fig. 6 | Projected contribution of each transformation towards the achievement of the SDGs in 2050.** Projections are for average progress made by each of the six transformations (T1 to T6) on the targets for each SDG and all SDGs combined (ALL). Grey bars show the projected progress for the baseline Build Back the Same Pathway or BBS. Coloured bars are calculated percentage point contributions for each transformation towards additional progress made in the Six Transformations Pathway (STP) (Methods: 'Assessing interlinkages').

in the case of T2 (economy) due to interactions with T4 (energy) where combining their interventions results in conflicts and dampening effects. For example, material efficiency targets and additional taxes (T2) slow economic production, industrial output, and emissions, while at the same time measures to decarbonise the energy system (T4) both reduce material intensity, productivity and emissions while also stimulating growth in green industrial output and material consumption. There may also be other areas of overlap in terms of the modelled impacts of individual transformations on the outcome targets which are not necessarily trade-offs, for example where the effects of supply- and demand-side measures in different transformations result in similar effects. The net positive result in T3 (food) reveals a multiplier effect where aggregate gains for STP are greater than the sum of individual gains (Fig. 5c). This boost in progress is unlocked when multiple interventions such as social protection (T2) and investment in sustainable agriculture (T3) leverage greater gains in food security and nutrition targets.

**The contribution of the six transformations to accelerating SDG progress**

Fig. 6 presents the simulated contribution of each transformation towards achieving the 17 SDGs in 2050 (Methods: 'Assessing interlinkages'), which reveals the complex interactions between the six transformations and SDGs. We choose to evaluate these effects in 2050 (as opposed to 2030) to capture the long-term implications of the transformations. Overall, T1 (wellbeing) makes the greatest contribution, particularly for goals relating to poverty (SDG1), food and nutrition (SDG2), economy (SDG8), cities (SDG11), and climate action (SDG13). This reveals the positive effect of increasing resilience (through investment in adaptation in T1) on the SDGs, particularly over the long-term. T6 (environment) also contributes strongly to SDGs progress, however, these gains are largely associated with land and biodiversity (SDG15) and oceans (SDG14) which accelerate from a very low baseline. T3 (food) mainly accelerates performance on food and nutrition (SDG2) and biodiversity (SDG15), with positive effects on many other goals. T4 (energy) makes a strong contribution to climate

action (SDG13) as well as energy (SDG7), economy (SDG8), industry (SDG9), and cities (SDG11). For T5 (urban) the biggest contributions are towards sustainable cities (SDG11), energy (SDG7) and climate action (SDG13). Finally, T2 (economy) results in moderate gains for a broad range of goals, primarily on poverty (SDG1), gender equity (SDG5), economy (SDG8), income inequality (SDG10), and government revenue (SDG17).

## Discussion

Global research has identified six critical transformations needed to achieve the SDGs by 2030[18,19]. Our study shows how the six transformations could be accelerated in a national context to advance progress towards the SDGs and contributes several important knowledge advancements. Firstly, by modelling all six transformations and all SDGs we provide a complete and holistic picture of the combined effects and complex interactions of these multi-system transformations and their interlinkages with the SDGs. Secondly, by modelling key policy packages or accelerators we provide insights to support decision makers seeking to advance more rapidly towards the SDGs by 2030, while ensuring that gains made are resilient over the longer-term through a net zero outcome. Thirdly, by combining modelling with socio-technical analysis and transformation storylines, we provide an analysis of transformation pathways which systematically diagnoses impediments and explains the actors and processes that create favourable conditions for the ambitious new policies to succeed. Through this approach, we provide insights for governments and other actors not only on what policies are needed but also on how transformations can be enabled and accelerated.

The modelling results confirm the critical role that governments play in accelerating the six transformations and progress on the SDGs. The mix of policies packaged for each transformation require only modest economic stimulus (+7.5% additional expenditure per annum to 2030) and result in substantial additional gains on the goals (+23 percentage points by 2030 or +34% by 2050 in STP compared to BBS) (Fig. 2). The modelled policy measures also provide guidance for national policymakers on the type and magnitude of measures needed to accelerate each transformation. For Australia, important policies include tax reforms and expenditure on social welfare and green manufacturing (T2), as well as targeted incentives and public investments in adaptation and resilience (T1), sustainable agriculture (T3), energy decarbonisation (T4), demand reduction and electrification in transport and the built environment (T5), and environmental protection and restoration (T6).

While our SDGs policies end in 2030, a continuation of additional expenditure could yield further long-term gains, particularly if targeted at transformations and goals that continue to lag behind. Priorities could include more transformative policies for a sustainable and just economy (T2), particularly where they deliver progress on SDG9 (sustainable industry), SDG5 (gender equality) and SDG12 (responsible consumption and production). However, closing the gap to 100% SDG achievement (even by 2050) may be impractical due to diminishing returns on investment and pervasive trade-offs that are difficult to resolve, as noted in other studies[14,15].

By evaluating synergies and trade-offs between the transformation policy packages and the SDGs, our study provides a link to policy and planning decisions as called for in the literature[23,78–84]. The modelling results (Figs. 5 and 6) show that all six transformations together deliver the strongest gains on the SDGs, and help to inform more coherent and systemic strategies that harness synergies and manage trade-offs.

To maximise opportunities for synergies, actors could prioritise transformations with the strongest positive spillovers as identified through our modelling (Fig. 5). Firstly, the rapid transition to renewable electricity (T4) has the largest positive spillover effects on all other transformations, by enabling investment in green manufacturing

and zero carbon industries which supports a sustainable economy (T2) and opening opportunities to decarbonise the built environment and transport systems which supports sustainable urban development (T5). Secondly, the additional investment in adaptation and resilience in the wellbeing transformation (T1) also has large positive spillovers for other transformations and generates strong synergies for SDGs progress over the longer-term (Fig. 6). Thirdly, transforming the food system (T3) has large positive spillovers for transforming the environmental commons (T6) and is critical to make gains on biodiversity targets. Finally, dual investments in social protection and livelihoods (T2) and a sustainable food transformation (T3) can unlock rare instances of multiplier effects.

In advancing the six transformations, policy makers also need to be cognisant of important trade-offs identified by the modelling, particularly with regard to the sustainable and just economy (T2) and energy decarbonisation (T4). Here, internal conflicts within T2 regarding the sustainable versus just objectives are evident, whereby efforts to deliver a green and resource efficient economy reduce industry output in some sectors with implications for income and employment. This could explain the more limited progress made towards the 2050 targets for this transformation. Complex feedbacks between T2 and T4 also result in potential trade-offs, whereby policies for material efficiency targets and additional taxes (included in T2) slow economic production, industrial output, and emissions, while at the same time policies to decarbonise the energy system (included in T4) both reduce material intensity, productivity and emissions while also stimulating growth in green industrial output and material consumption. Interactions between these policies result in trade-offs which can constrain progress on the SDGs.

The socio-technical analysis underscores that the ambitious policy action incorporated in our modelling is not guaranteed and will face many impediments which are largely associated with technological, social, and political factors. Across the six transformations, common impediments include large sunk investments in unsustainable infrastructure or industries and market concentration in many sectors which creates resistance from powerful vested interests, policy settings which favour market incumbents, and societal pushback to required changes in lifestyles and practices. Actors seeking to accelerate transformations will also need to consider these impediments and ways to overcome them. As described in our transformation storylines, these enabling conditions include emerging crises, shifts in narratives and public opinion, maturation of innovations, new coalitions and social movements, and support from powerful actors.

We also acknowledge a range of limitations and important caveats in interpreting the study results and their broader relevance for other countries. Firstly, uncertainty in long-term modelling propagates in the model structure and data, parameterisation of model assumptions, and future global conditions[85,86]. Our sensitivity analysis reveals that some of the gains made in the STP would be lost due to worsening global conditions. However, this does not minimise the need for national strategies and actions, and crises can also trigger transformative change. Secondly, while synergies appear to dominate in our analysis, the results potentially gloss over localised trade-offs that occur from structural changes in the economy and create winners and losers in different industries and localities. The national scale of the modelling considers aggregate rather than localised effects. Potential areas for further research could include using scenario discovery approaches to support identification of the most robust combination of interventions[87], spatially downscaling the modelling framework to consider localised effects, and endogenizing socio-political factors and tipping points to provide a more complete quantitative analysis of transformation pathways[45,88].

## Methods

The methodological framework for our analysis (Fig. 1) is informed by the six transformation entry points[20] that have been defined as imperative to SDGs achievement, and by recent advancements[25,27] that bridge scenario modelling and transition studies to provide a more comprehensive analysis of transformation pathways. This approach integrates important techno-economic considerations through policy- and technology-rich modelling with broader social and political developments using socio-technical analysis and transition storylines. As such, it addresses issues that are often excluded from modelling studies and enriches the analysis with empirical findings from socio-technical research[25,32]. Building on the steps described in the introduction, a more detailed description of the study methods is provided here.

### Critical juncture and pathway metanarratives

Advancing on our previous research[15] which used a conventional scenario matrix approach nested within the global SSPs, this study starts by describing two alternative future pathways which are separated by a critical juncture resulting from compounding crises. Such junctures are a well-documented feature in transformations, and are often associated with major crises or shocks which destabilize the existing regime and open windows-of-opportunity for change[39,40]. An important premise for our analysis is that several cascading crises create the potential for a critical juncture or bifurcation, with two emerging pathways. These pathways are equally plausible and are guided by metanarratives that broadly align with Business-as-Usual and Sustainability Transition scenarios for Australia[15]. They also draw exogenous global assumptions of landscape trends where relevant from SSP2 (Middle of the Road) and SSP1 (Sustainability)[89] as well as modelling from other international organisations and research institutes, scientific literature, and national studies in Australia (Supplementary Table 2). These include exogenous parameters associated with population (net migration), governance, global temperature changes and impacts, climate change adaptation costs, demand for commodities, interest rates on debt, and average technology improvements.

The Build Back the Same Pathway (BBS) assumes that during Australia's medium-term recovery from 2021 to 2030, policy settings and ambition return to pre-COVID levels with no additional investment beyond the emergency stimulus measures (Table 1). This places Australia largely on a business-as-usual recovery pathway which sees a continuation of pre-COVID trends. Exogenous global assumptions (Supplementary Table 1) are based on SSP2[90] and the BAU scenario from Allen, Metternicht[15].

Contrasting this, the Six Transformations Pathway (STP) (Table 1) assumes that a pervasive narrative emerges in Australia around the need for structural change and to build back better using the SDGs as a roadmap. This gains support from powerful actors and coalitions which legitimizes stronger policy action. Central to this action is an extension of economic stimulus over the period from 2021 to 2030 which places Australia on an accelerated pathway towards the SDGs and longer-term transformations. Exogenous global assumptions (Supplementary Table 1) draw from SSP1[90] and the Sustainability Transition scenario from Allen, Metternicht[15].

Both pathways incorporate the measurable effects of COVID-19 and other shocks as well as the government's emergency response and stimulus measures taken largely in 2020/21 (Supplementary Fig. 1). Following the critical juncture the pathways diverge considerably from 2021/2 with alternative medium-term recoveries which place Australia on considerably different pathways.

### Literature review and socio-technical analysis

Nested within the broad metanarrative for the STP we developed six transformations (Ts) encompassing the full complement of systems to

be transformed to achieve the SDGs, as articulated in global studies[18,19]. Secondly, for each T we reviewed empirical research, policies, technical reports and analyses, government data and other literature to provide a socio-technical analysis of current regime conditions, impediments, policies, and promising niche innovations. The development of the standardised template used to synthesise socio-technical insights for each T (Supplementary Table 2) was informed by our comprehensive review[24] of a large and growing body of research on sustainability transitions and transformations which explores common impediments and enabling conditions for accelerating transformations[5,7,40,43,46,53]. The template includes a review of recent progress and regime challenges including common types of impediments identified in the literature (techno-economic, social-behavioural, institutional-political, and social-ecological lock-ins) which lead to system inertia and path dependence[27,46,52,53,69]. It also identifies emerging innovations and initiatives which provide opportunities or positive seeds[91] for transformative change.

A range of context-specific source material was reviewed by the authors to populate the templates, including empirical research on sustainability transitions in Australia, national data and existing scenario modelling and foresight studies, assessments of Australia's progress on the SDGs, government strategies and reports, and analyses and technical reports from think tanks, the private sector and civil society organisations (Supplementary Table 2). Relevant literature was identified through online queries (e.g. Google Scholar) for academic and grey literature specifically pertaining to Australia as well as through Web of Science queries used for our review of academic literature on accelerating transformations[24]. This analysis is then used to both inform the selection and parameterisation of quantitative policies as inputs to the modelling, as well as the development of the qualitative transformation storylines which explain the conditions needed for the policies to succeed. Given the very broad scope of the analysis covering a comprehensive suite of transformations, the intention was to build on existing research to develop a mix of key interventions needed to accelerate progress along with explanatory storylines addressing the enabling conditions and context needed to support the quantitative pathways.

## Elaboration of quantitative policies and transformation storylines

Each of the six transformations comprises a range of different quantitative policy interventions as well as a qualitative transformation storyline. This creates a bridge between the quantitative model projections which are largely techno-economic and the broader socio-political enabling conditions and actor strategies needed to overcome key impediments. These are informed by the socio-technical analysis for each transformation which addresses the co-evolution of techno-economic and socio-political dimensions, including common impediments associated with incumbent actors and institutions and engrained norms and behaviours as well as emerging niche-innovations which provide promising seeds of change.

Quantitative settings include a range of measures in the *iSDG-Australia* model including government expenditure, taxes, subsidies, policies as well as assumptions regarding technology diffusion and behavioural changes (Supplementary Table 3). Tax, subsidy and expenditure settings for the BBS remained in line with recent time series data (as a proportion of GDP), while other assumptions were informed by time series trends and existing research. In the STP, each transformation includes an ambitious package of SDGs policies from 2021 to 2030, beyond which policy settings return to trend except for net zero policy assumptions which are continued to 2050. The SDGs policy stimulus is intended to generate a period of acceleration to 2030, while longer-term settings ensure consistency with achieving net zero targets. Policy settings and assumptions for each transformation were parameterised based on official time series datasets,

available costings studies and reports, other modelling studies and research for Australia, global benchmarking (e.g. against OECD countries), and iterative analysis of model projections (Supplementary Table 3).

Complementing this, the transformation storyline explains the processes and mechanisms that create favourable conditions for the successful implementation of the ambitious new policies and interventions which accelerate each transformation. A brief synopsis of the storyline for each transformation is provided in Table 2 (see also Supplementary Table 2). This incorporates insights from the socio-technical analysis on key impediments and emerging opportunities and seeds of change associated with each transformation. These impediments vary between systems and can result from large sunk investments which create vested interests, economies of scale which challenge new market entrants, lifestyles which become organised around unsustainable practices, and policy settings and networks which favour incumbents or stifle innovation. Policymakers can become captured by vested interests, tied up by lobby groups, or lack the capacity, resources and incentives to act[5,92]. Important conditions for overcoming inertia result from a range of sources and societal actors. These include changing external pressures from the maturation of emerging innovations which provide solutions that policymakers can push, shifts in public opinion and pervasive narratives, coalitions that organise actors towards new goals, and support from powerful actors and policy entrepreneurs[7,46,93,94]. Shocks, crises and slow-moving trends can generate instability in existing systems, creating windows of opportunity for systems change[39,40,42].

## Model description, calibration and validation

The study applied an integrated, macroeconomic system dynamics model (*iSDG-Australia 2.0*), the original version of which is detailed in Allen, Metternicht[15]. The *iSDG* family of models[30] are built in a stock and flow structure and formulated as a set of differential equations encompassing 3000+ variables organised across 30 economic, social and environmental sectoral modules. A description of each of the sectoral modules along with key assumptions and source literature is available in the model documentation[30] and a brief summary of the model structure is provided in Supplementary Fig. 8.

The latest version of the *iSDG-Australia* model (v2.0) incorporates an expanded set of 37 modules which include recent advancements for Australia's built environment[95] and transport sectors[96,97], amongst others. For this study, we further develop the model to incorporate a COVID-19 shock in key economic and population sectors and the associated government emergency stimulus response over the period 2020-21. A transformations module was also developed and structured around the six transformations to achieve the SDGs. This included both a set of interventions for accelerating each transformation as well as a suite of transformation targets for measuring progress on each transformation over the period to 2050. Enhancements were also made to a range of sectoral modules to increase the suite of policy measures and assumptions available for each transformation, including for fuel switching, green manufacturing, accelerated phase out of technologies, sustainable agriculture, and waste management.

Drawing on good practice model verification procedures[98–102], model validation included both structural and behavioural validation. The model is calibrated on an extensive database of 25–30 years of historic time series data commencing in 1990 and sourced from official and verified national government sources (Australian Bureau of Statistics and government administrative databases), as well as official data from international databases (Supplementary Tables 3 and 4). In each sector, parameters were calibrated using multi-parametrical optimisation and reference parameter ranges. Behaviour reproduction tests were used to evaluate the goodness-of-fit of simulated and actual data using plotted graphs and error statistics ($R^2$), mean percentage error (MPE), and mean absolute percentage error (MAPE)[100].

Goodness-of-fit statistics calculated for a selection of critical variables for the baseline BBS simulation are included in Supplementary Table 5. Comparisons of baseline projections and data for a broad selection of economic, social and environmental variables are provided in Supplementary Figs. 9 to 11.

## Model projections

Following final calibration of the *iSDG-Australia 2.0* model, the baseline BBS was projected through to 2050 based on a continuation of current policy and expenditure settings. Parameterisation for each transformation in the alternative STP was based on the settings in Supplementary Table 3. The simulation period was set to include the implementation period for the SDGs as well as net zero (2021–2050), with alternative policy assumptions introduced from 2021 onwards and with most ending in 2030. In some cases, the model uses time delays which result in more gradual effects from interventions.

To explore individual and aggregate effects of the six transformations, each transformation was projected individually before projecting all transformations simultaneously as an aggregated STP. This enabled an assessment of interactions and potential spill-over effects between the different transformations, for example the positive effects that a transformation towards a sustainable and just economy (T2) might have on transforming wellbeing and capabilities (T1). It also enabled an evaluation of the impacts of each transformation on the full suite of SDG targets and indicators, as well as a comparison of individual and aggregate results.

## Targets for 2030 and 2050 and method for assessing progress

The performance of the BBS and STP as well as each of the six transformations were evaluated against a set of 80 unique indicators covering all 17 goals in 2030 and 2050, as well as a subset of 67 indicators or transformation targets (TTs) covering each of the six transformations in 2050 (Supplementary Table 4). The selection and prioritisation of SDG targets and indicators was based on the official SDG indicator framework[103] as well as recent baseline assessments of Australia's progress on the SDGs[33,104]. The 80 indicators correspond to all 17 goals and 52 targets. Target values for each indicator for 2030 and 2050 were formulated drawing on a range of sources, including the SDG targets themselves, additional targets used in Australia's SDG baseline assessments and previous modelling[15,33,104], threshold values taken from the global SDG Index[2] and other global studies and benchmarks (Supplementary Table 4). The aim was to formulate ambitious but credible targets for Australia to reach by 2030 and 2050, however, in most cases they are not official Australian government targets. Across the 80 SDGs indicators included in the model, we classify 23 as economic, 29 as social and 28 as environmental which provides comparable representation of the three dimensions of sustainable development.

The 67 TTs are a subset of the 2050 SDGs targets and are used to evaluate progress made on each transformation and explore acceleration dynamics and spillover effects. Each transformation is allocated a unique set of TTs based on thematic relevance (see Supplementary Table 4) and the number of targets varies between the different transformations (Table 2). We avoid duplication of TTs across the different transformations and exclude means of implementation indicators (from SDGs 16 and 17) which could apply to all transformations. Including these as TTs for all Ts would have made interpretation of the results less clear. Additional SDG indicators were excluded where they were considered duplicative or did not align with the transformation storyline. The allocation of targets was also informed by global studies which provide guidance on 2050 targets[17] as well as on linking SDG indicators to the six transformations[105].

Progress towards each target is simulated in the model over the period from 2016 to 2030 (for 2030 targets), and 2031 to 2050 (for 2050 targets). The projections reveal Australia's proportional achievement of a target (from 0 to 100%). A normalised scale (0–100) was used, whereby the reference value in 2015 was considered the zero point and the target values for 2030 and 2050 were considered the final points (reflecting % progress). For the 2050 targets, the projected performance by 2030 was used as the starting point for measuring additional progress to 2050. Average performance on the SDGs targets is aggregated firstly at the SDG target (for 52 targets) and then goal level (for 17 SDGs) so that each goal contributes equally to the overall SDGs performance regardless of the distribution of indicators. Similarly, average performance on the TTs is aggregated at the transformation level so that each of the six transformations contributes equally to average overall performance. We acknowledge that the averaging or aggregation of indicator performance at the goal level or for all goals glosses over contextual information on performance of specific indicators. However, it was necessary in our study for pragmatic reasons to present and discuss the study results for such a large set of indicators.

## Assessing interlinkages

Interlinkages between the different transformations as well as between the transformations and the SDGs are explored in both a qualitative sense (drawing upon important interlinkages highlighted in the storylines) as well as quantitatively through the modelling results. The very broad scope of the system dynamics model combined with the six transformations approach supports a complex quantitative analysis of feedbacks and interlinkages across different systems and targets. Each transformation has a unique set of TTs which we use to evaluate progress on each transformation by 2050. In STP, we simulate all six transformations simultaneously and as such the projections represent their aggregate effects. While each transformation is designed to accelerate progress towards its own unique set of TTs, it also has implications (synergies and trade-offs) for the achievement of TTs associated with the other transformations.

We explore these interactions by simulating each transformation individually and evaluating effects on the total set of TTs as well as the TTs associated with each transformation (Fig. 5). Synergies (trade-offs) occur when the simulation of a transformation results in improved (worsened) performance on TTs associated with other transformations when compared against the BBS baseline. We then sum these individual results and subtract them from the results from the STP when all transformations are simulated simultaneously to highlight any discrepancies. Where these values are negative, they suggest aggregation losses involving potential areas of duplication or hidden trade-offs when transformations are combined. Where they are positive, they suggest multiplier effects where additional gains are only made when transformations are combined. We interpret synergies in a broad sense, including positive spillovers between the transformations as well as stronger multiplier effects.

In a similar way, we also explore the effects of each transformation on the achievement of each of the 17 SDGs. As we are interested in the long-term effects of each transformation, the analysis is done using the projected results for 2050. We calculate the additional percentage contribution that each individual transformation makes towards the achievement of each SDG in 2050 when compared against BBS baseline. To present these results in Fig. 6, we convert these values to a proportional contribution by dividing the individual values by the sum of additional percentages from all transformations on each SDG. We then multiply these proportional contributions by the results for the STP (i.e. by the projected additional gains on each SDG made by the STP compared against the BBS). This enables us to decompose the results for the STP and allocate a contribution for each transformation towards additional progress made on the SDGs.

## Sensitivity analysis

Sensitivity analysis was used to complement the model calibration process and to test key model assumptions. For very large system

**Table 2 | Brief overview of the six transformations storylines and associated TTs (see also Supplementary Tables 2 and 4 for the full storylines and list of targets)**

| STORYLINE | TTs |
|---|---|
| **T1. Human wellbeing and capabilities** | |
| Following COVID-19, governments and societal actors commit to upgrading Australia's health and education systems to build back better and ensure that Australia is well-placed to respond to future shocks. Acute public awareness of the health and education system failings and growing concerns around natural disasters builds public support and momentum for increased investment in resilience and health and education systems and reforms. This is supported by changes in the way that public expenditure is screened and allocated based on wellbeing. The adoption of systems approaches leads to the development of a National Preventative Health Strategy which effectively brings together partners across all levels of government and healthcare providers, professional associations, industry, NGOs, First Nations groups, and individuals. This builds momentum for change, shifts the narrative towards preventative health, and builds public support for new investment and reforms. The rapid emergence and scale-up of digital technologies provide greater accessibility to services. | 16 targets relating to education, health and resilience. |
| **T2. Sustainable and just economy** | |
| Multiple crises bring the rising cost of living pressures, government debt, stubborn poverty rates, and rising inequality into sharp focus with increased media coverage raising public awareness and community support for action. This creates the burning platform needed to pressure governments and unite stakeholders to embrace much-needed tax reform and to consider new ways of prioritising government investments. Successes in neighbouring and like-minded countries encourages well-being initiatives in Australia, leading the federal and state governments to adopt wellbeing budgets to screen major public expenditure. Tax reforms provide finance to support all six transformations, including increased social transfers and new investment in infrastructure for green hydrogen and manufacturing industries. Public and government pressure aligns private capital with wellbeing objectives and the SDGs along with divestment from unsustainable industries. This builds momentum over time for economy-wide regulations and standards which place stricter controls on pollution and emissions. | 21 targets relating to sustainable consumption and production, green industry, and jobs and social protection. |
| **T3. Sustainable food systems** | |
| The bushfire devastation, unprecedented floods and COVID-19 shine a bright light on the extreme shortcomings in Australia's food system. A regenerative agriculture movement gains momentum with impetus from popular books and films and support from powerful actors. Emerging business models such as farm to table distributors, the proliferation of local farmers markets and changing preferences for healthy diets and organic produce support momentum for change. Many emerging technologies begin to disrupt the food system and provide viable alternatives that are pushed by governments, business and civil society. Shifting narratives and values around healthy diets and lifestyles begin to erode support for current incumbent firms, with people seeking out local farmers markets and delivery alternatives. Over time, governments and stakeholders reach a shared agreement on the desired characteristics of a regenerative future food system. Governments provide new incentives, extension services and financing options to support farmers to adopt regenerative practices leading ever-greater numbers of farmers practicing agroecology over ever-larger territories and which engages more people in the processing, distribution, and consumption of agroecologically produced food. | 8 targets relating to sustainable food systems and nutrition. |
| **T4. Energy decarbonization** | |
| Following the Black Summer bushfires and unprecedented floods, public support for action on climate change reaches new levels and powerful actors call for a green recovery from COVID-19. Bottom-up political movements and collective action see a shift in politics away from the status quo in support of decisive policy on climate change, disrupting incumbents and providing a window of opportunity to end the climate wars. A powerful coalition of politicians, business, community and unions agrees on shared ambitious mitigation targets for Australia, supported by a clear plan for investments needed over the next 10 years to accelerate the transition towards 100% renewables. Longer-term plans are developed to reduce demand and tackle emissions in hard-to-abate sectors. Stakeholder activism and divestment and hostile takeover of fossil fuel assets by powerful actors result in an accelerated phase out of fossil fuel generation. Investment in R&D results in continued technology advancements which provide solutions that policy makers can push over the longer-term to support net-zero shifts in long-haul transport, agriculture, and industry. | 3 targets relating to energy access and decarbonisation. |
| **T5. Urban development** | |
| As the homes of many economic and cultural leaders and powerful actors, cities set about driving changes to corporate behaviour and turning up the heat on state and federal governments. Building on local initiatives, a national framework of local visions and plans are developed and tailored for each city including ambitious goals and targets aligned with the SDGs. This improves community and sectoral buy-in, guiding policy measures, generating investments, and raising awareness. Targets and plans support rapid decarbonisation over the next few decades and coherent policies across sectors backed by investment and incentives from all levels of government in social housing, electrification of buildings, circular economy and waste reduction, local food systems, behavioural change towards sustainable diets and lifestyles, and the electrification of transport and charging infrastructure. | 10 targets relating to transport, built environment, water and sanitation and waste. |
| **T6. Environmental commons** | |
| Through the COVID-19 lockdowns, an increasing appreciation for nature emerges as people seek the great outdoors for relaxation and recreation and thousands relocate from major cities to regional areas. Building on the experience in the latest State of Environment report, a more holistic understanding of Australia's environment is enabled which feeds through to new partnerships to manage Australia's natural assets. Connections between people and country, between the economy and the environment, and between western scientific and Indigenous knowledge systems begin to flourish, with stakeholders in government, business, research, and civil society working together to deepen these connections and build a shared vision for a nature-positive society and economy, guided by science-based targets aligned with the SDGs and other global frameworks. This is supported by transformations in food systems, dominant patterns of production and consumption, energy decarbonisation, and urban systems. | 9 targets relating to land, marine, freshwater and climate. |

dynamics models such as the *iSDG* tool, good practice is to focus on those relationships and parameters that are both highly uncertain and likely to be influential[100]. Previous sensitivity analysis for the *iSDG* model has found that sensitivity of the simulated SDG performance results for baseline and alternative scenarios to varying global assumptions is low to moderate[15,95].

The national scope of the model does not enable the inclusion of endogenous structure for global drivers such as trade, interest rates, action on climate change, or migration. We tested the sensitivity of model outputs to ten key exogenous global assumptions (Supplementary Table 6). The output variables of interest included the progress across each of the six transformations, performance scores for each of the 17 goals, aggregate transformation and goal performance, and population, real GDP and GHG emissions.

We followed the general workflow described in Pianosi, Beven[106], running Monte Carlo simulations in which model parameters were randomly adjusted within a predetermined range (min/max) using a normal distribution. We adopted the all-at-a-time (6000 simulations) random Latin Hypercube sampling method. Input ranges for each sensitivity variable were informed by available literature and previous modelling studies (Supplementary Table 6).

## Data availability
The model input dataset used for calibration as well as model projections to 2050 generated during this study are available in the Supplementary Data 1 which is deposited on the figshare repository (https://doi.org/10.6084/m9.figshare.22815317). Additional materials, charts, tables, and data are available in the Supplementary Information.

## Code availability
The *iSDG* simulation model is owned by the Millennium Institute and is subject to third-party restrictions. The model can be made available from the Millennium Institute for research purposes on request.

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

## Author contributions

C.A., A.B. and T.W. conceived the study and all authors contributed to the aspects of the study design; M.P. and C.A. developed the iSDG-Australia model; C.A. and A.B. developed new model extensions and were responsible for data collection; C.A. led the drafting of the paper, model simulations, data analysis and visualisation of results with contributions from A.B.; T.W., M.P. and S.M. contributed to the drafting and review of the paper. All authors edited and approved the final manuscript.

## Competing interests

The authors declare no competing interests.
