## [Peer Review File · Nature Communications]

Modelling six sustainable development transformations in Australia and their accelerators, impediments, enablers, and interlinkagesREVIEWER COMMENTS

Reviewer #1 (Remarks to the Author):

Modelling six sustainable development transformations and their accelerators, impediments, enablers, and interlinkages

This manuscript presents a national scenario analysis of the progress towards sustainable development goals (SDGs) in Australia, based on a simulation model called iSDG. The text is well-written, and the figures and tables, summarizing an extensive analysis, are clear. I congratulate the authors for this systematic analysis. However, I have three major concerns about this study:

1) The manuscript lacks a clear objective and analytical focus. In Line 48-51, the authors say that they make a significant advancement to national modelling studies to address the prespecified research gaps, and focus on improving the understanding of accelerators and impediments. While acknowledging that the subject of SDG achievement is immensely broad and complex, I still find the objective statement unfocused and overly ambitious, also considering that the previous paragraph on the research gaps is not fully convincing either. Line 40 says that modelling studies have not so far included six transformations simultaneously, but there are many studies that take interactions of SDG transformations simultaneously, as the authors cite, therefore the reader wonders which specific SDGs or transformations are missing in those existing studies and what this study adds. Without such a clear objective, the extensive analysis presented in this manuscript seems dispersed.

2) The manuscript does not employ a clear framework for model-based policy analysis, hence results in a confusion of terminology. For instance, from Figure 1 and the corresponding text, I understand that the authors use Transformation Pathway (TP) to refer to policy scenarios, and they analyze their impact on SDG-like targets, which are policy objectives. Then, there are 'accelerators' which are apparently also policy levers, and the figures show the impact of TPs on TPs (e.g. Figure 3 & 5). I suggest the authors to use a well-defined policy and scenario analysis framework, for instance similar to the XLRM framework developed at RAND Corporation or the good old system analysis framework described in Miser and Quade (1984) Handbook of Systems Analysis.

3) Transparency is a main issue both in the manuscript and the underlying analysis. The model and data used in this analysis is "available upon reasonable request", which is not commonly seen anymore in scientific studies. There is no description in the manuscript about the core components and assumptions of this model, and how they relate to the six transformations and their interlinkages analyzed in this study. A summary, at least, would have been useful. The model documentation cited in the bibliography tells us only the relationships, not how they are formulated. Furthermore, the validation and calibration procedure is described in the Methods, some performance metrics are presented in the supplementary material, yet there is no clear visualization of these validation results, whether the model replicates the historical data, at least. Considering that the manuscript presents this analysis as a decision support tool with almost point estimates, not for what-if analysis, the reliability of the model seems to be even more important. In the current manuscript, however, the reader is left with a black box.

Relatively minor comments:

- Line 81-83 : I doubt that the terminology of V-shaped and A-shaped dynamics is commonly used. Could you explain this further?
- Figure 2 (and the other figures) presents the projections for aggregate SDG progress, which is the average percentage of progress towards each goal. While such aggregate metrics, such as the SDG index of SDSN, is commonly and unavoidably used, it bears the drawback of over-aggregation and losing the contextual information and insights. The motivation for choosing such aggregate metrics, and the limitations of it, is not discussed.
- In Figure 2 and others, sensitivity analysis is based on 10 uncertain parameters described in the SI.

In such a large model, I find it hard to believe that there are only 10 specific uncertain parameters. This relates to the opacity of model structure and assumptions. Could you please elaborate on this?

- In Figure 3 and the corresponding discussion of the results, the authors refer to the social tipping concept and state that SWT trajectory results in an accelerated transition. Regarding the tipping concept, I recommend a more careful use of the term (<https://wires.onlinelibrary.wiley.com/doi/full/10.1002/wcc.813>). Regarding the results in Figure 3, I see that almost all transformation targets show an S-shaped trajectory also in the BBS case, and they have already demonstrated acceleration, maybe even passed the inflection point. SWT seems only to shift the saturation point up. Please elaborate and describe the results accordingly. Also, given that the reader does not know about the model structure, it is not clear if these S-shaped dynamics are created by endogenous model structure, or embedded in the model.
- In Figure 3, projections start from 2016, which is quite in the past and data must be available for more recent years. Could you explain what motivates you not to start the model from a date closer to the present? This also is not consistent with other figures where projections start in 2020.
- The storylines underlying the transition pathways are described clearly in detail, yet not how they were developed. In the Methods, I read that the authors developed the storylines with desk research based on the relevant literature. In the SI, I see common dimensions included in each description. It would have been useful to develop those with a robust framework for literature review and storyline components, or to describe such a framework if it used. Additionally, Lines 527-539 are not describing the methods, but the content.
- I am skeptical about the methodological choice of summing up the fractional effects as synergies on the last column of Figure 5a, which is not meaningful metric. Please motivate this choice.
- Throughout the manuscript, the text is not very clear which arguments are based on the findings, and which are the interpretation and reflection of the authors based on the literature. For instance, Lines 405-411. I suggest a clearer distinction between model findings, TP narratives, and the broader discussion.
- In the SI, I suggest having Table 3 before Table 2, since the former has the narratives and the latter has the quantification of those.

Reviewer #2 (Remarks to the Author):

Thanks for inviting me to review this interesting paper. The study uses a system-dynamics model to assess progress towards the SDGs in Australia, both for a trends-continued and an ambitious transformation scenario. I see three important contributions in the paper: i) linking qualitative storylines of different transformations with quantitative modelling, ii) providing a breakdown how different individual transformations contribute to the overall transformation pathway, and how they interact, iii) describing many of the interventions in the individual transformations with explicit policies (e.g. increasing education expenditure, or increased tax rates for higher incomes), as opposed to reflecting them just via their desired outcome.

However, I also have a number of questions and recommendations, some of them conceptual and some on modelling choices and results. If these can be addressed by the authors, the paper would certainly make an important contribution to the literature on transformation pathways towards the SDGs.

Conceptual points:

1) I found the terminology sometimes a bit ambiguous and confusing: "Transformation pathway (TP)" seems to refer to multiple things:

- the scenario resulting from implementing a single transformation
- the set of interventions or levers (reflected through changing parameter settings in the model) associated with a single transformation

- progress in a set of outcome indicators associated mainly with a single transformation, but when all transformations are applied (e.g. Fig. 3A)

Here it would help the reader if these related but different concepts could be separated more clearly. Besides that, recognizing that the transformations are not fully independent, I am not sure if TP is a good wording for single transformations – in the literature it seems to be used more commonly for referring to the overall transformation.

2) The authors have done a great job in documenting the narratives, modelling choices and indicators associated with each of the six transformation in the Supplement. However, for the reader of the main manuscript things are not always immediately clear. I would find it helpful to briefly summarize the transformations (e.g. very short qualitative description, selection of key modelling choices & key indicators) in a short table in the main manuscript.

Related to that, the discussion in the Results section focusses mainly on TP 2-4. Why was this choice made?

3) I am generally a bit skeptical about averaging together the breadth of the SDGs into single aggregate indicators of progress. In my view, such aggregate indicators lump together so many different things that important details can get lost. (Although I do acknowledge that this is widely done in metrics such as the SDG index).

However, my main concern related to such aggregate indices is that they can be prone to weighting biases, such as: i) the SDGs contain more human-development related goals and indicators than environmental ones, which makes it easier for high-income countries (high human development, but also high environmental pressures) to score high. ii) Models are incomplete in their coverage of indicators, and tend to cover some SDGs better than others. This potentially introduces a second weighting issue.

To overcome this, I would recommend two things:

a) in addition to the single aggregate metric, split it into a number of broad categories (could be just human development vs environment, or along the “five P” clusters of SDGs.)

b) make it more transparent how much individual SDGs contribute to the overall index, e.g. by adding the number of indicators for each SDG to Fig. 2A.

4) The S-shaped acceleration curves hypothesis seems to rest largely on Figure 3. However, if I understand correctly, the “emergence phase” shown in panel A is not based on data but just an interpolation. Here I would strongly recommend to plot actual historical data, as I suspect the historic trends could look quite different from the suggested emergence phase. For some indicators, such as the environmental ones, it would perhaps not even show an emergence phase, but a worsening until a turnover point. [If it turns out to be impossible to obtain historic data for all required indicators, using the calibrated model for backcasting would be OK as a fallback].

In light of the updated Figure 3, it might be required to nuance the hypothesis of S-shaped acceleration.

5a) I found the discussion of the interactions of transformations around Figure 5 a bit confusing. This is in part related to terminology (comment 1 above), but perhaps also the concepts behind the given numbers can be explained a bit more clearly. For example:

- Generally: I understand that many of the stated “%” numbers are changes/differences in the relative progress metric (which is measured in %)? Then I would recommend to use “percentage points” as a unit whenever referring to differences.

- Related to “Synergies” column in Fig. 5A: I assume this is the row sum excluding the diagonal, correct? Is this straight sum really meaningful? Instead I would recommend to evaluate how transformation 1 improves indicators from transformations 2-6, and so on. Due to different numbers of indicators, this can differ from the straight sum.

- Also, the wording with "synergies" is a bit confusing in the caption – it can be misunderstood as the "Synergies" column only counting positive interactions which are previously labelled "synergies".

- I suggest to add something like "all targets" to the header of Fig. 5B, to make the difference between panel A (transformation 1 on targets from individual transformations) and panel B (transformation 1 on all targets) immediately clear.

5b) On "aggregation gains/losses" as in Fig. 5C and associated discussion. What you call "aggregation losses" is, I think, in part caused by the fact that for interacting transformations the order of "applying" transformations in the decomposition matters.

For example, in order to reduce emissions from electricity generation, one can make changes on the demand side (e.g. more efficient appliances), or on the supply side (decarbonize power generation). If the demand transformation is "applied first", it has a fairly large effect on emissions (as every saved kWh is counted with a high emission factor). On the other hand, if the supply-side transformation is "applied first", emission factors are already low so the demand transformation has a seemingly small effect on reducing emissions. This does, however, not mean, that the two transformations (supply / demand), involve areas of duplication (only in the sense that they both target emission reductions) or even trade-offs.

Now this is of course an artificial example as only in models we can do such a decomposition analysis. In the real world these two transformation would happen simultaneously – and here I would consider the demand transformation as synergistic with the supply transformation (lower demand facilitates decarbonizing supply).

Perhaps the quantification and discussion can be adjusted so that it gives a clearer picture of what are actual trade-offs?

Modelling choices and results:

6) Australia-specific point: As noted in the narrative of transformation 4, Australia is currently the largest (or close second largest) exporter of coal worldwide. So a holistic SDG pathway for Australia has to include a phase-out of coal mining and exporting, at least in line with the global trend in phasing out these fuels, or perhaps even faster.

From the SI I gather that a 40% reduction in coal mining and 30% for oil and gas by 2050 is assumed in the transformation scenario, is that right? This is clearly not in line with the Paris agreement, for this a much faster phase-out would be required (e.g. for coal a phase-out by 2030 in OECD and around 2040 in non-OECD). The SSP paper by Riahi et al. is cited here as a reference – are these numbers taken from an SSP1 baseline scenario? If yes: a climate policy scenario (e.g. SSP1-RCP2.6) would have to be used instead, otherwise SDG 13 is not achieved.

7) Temperature change and climate impacts: Which SSP scenario is the ~1.8°C increase in GMT for the SWT scenario taken from?

Related: How are climate impacts considered in the model? I would expect climate impacts in Australia to amplify further from the current warming level to the ones assumed by 2050 (1.8°C – 2°C), so more wildfires, more droughts and rainfall extremes (even if national average precipitation stays constant as assumed).

I acknowledge that fully covering these impacts is challenging, but I would ask the authors to at least make a bit clearer to which extent they are covered, and where they see risks of the remaining impacts to jeopardize SDG progress.

8) The result that carbon dioxide removal covers around 30% of the required emission reductions towards net-zero (Fig. 4) is a bit surprising. Given the known trade-offs of large-scale CDR it should be checked if this compatible with the rest of the SDG agenda. Is this all happening through afforestation (transformation 6), or also BECCS? What is the combined land area used for land-based CDR? Are

side-effects on other SDGs (e.g. through land competition, water requirements, etc.) taken into account?

Also, related to point 7: Considering recent fires, and given amplifying climate impacts, I have some doubts if afforestation can really provide this amount of (long-term, stable) CDR in Australia. So perhaps the mitigation strategy needs to be adjusted, shifting more towards actual emission reductions on the energy side, and less reliance on CDR.

9) Overall progress in the aggregate SDG achievement indicator slows down remarkably after 2030, only moving from 82% to 89% over 20 further years (compared to ~40% to 82% in the 2020-2030 decade). Why is this? Additional policies (or a strengthening of the already used policies) after 2030 could be considered.

10) I compliment the authors for doing such a broad sensitivity analysis across multiple uncertainty dimensions. However, I would still hesitate to call the outcome "confidence bounds" and attach a probabilistic interpretation to them, for the following reasons: I understand that the variation of the input parameters reflects a range specified by the modeller, and not their true probability distribution (which is not known). Also, there are still uncertainties not considered (e.g. structural uncertainty related to the way dynamics are represented in the model). Perhaps "distribution of sensitivity runs" (or similar) described better what these bands represent.

Related, on the results of the sensitivity analysis: It is interesting that the distribution of sensitivity runs extends almost exclusively to lower SDG achievements than the main SWT scenario. Can this be interpreted as saying that there is a substantial risk of much lower SDG achievement even if all the transformations of the SWT scenario take place? Or is it more a feature of how the sensitivity analysis and/or model was parameterized?

11) The large dip in poverty rates in the COVID years (Suppl. Fig. 4, 2a) is surprising, given that the change in inequality (2b) is much more moderate.

Other points:

12) I would suggest to include "Australia" in the title to make it clear that it is a country-specific and not a global study.

Reviewer #3 (Remarks to the Author):

Reviewer comments to author:

The article presents a timely and impressive research effort that takes on a challenging analysis of how six transformative pathways toward sustainability could be materialized in the Australian context. It investigates how far these pathways – that in previous literature have been defined as imperative to SDG achievement – can push progress on the 17 SDGs by both the Agenda2030 'deadline' of 2030 and by a more long-term vision of 2050.

Overall, the paper is well written and easy to read. The paper furthermore seems to be based on robust and systematic qualitative and quantitative modelling. However, to make this article acceptable for publication, the latter point here must be made clearer, and weaknesses of the approach acknowledged. This can partly be addressed in the method chapter, but primarily it needs to be concisely described in the introduction chapter and highlighted in the discussion.

The type of system dynamics modelling that is undertaken is appropriate for the research question and powerful for exploring system behaviour in transformation. However, to its nature, SD models also tends to become a 'black box' for anyone on the 'outside', where assumptions on (future) system behaviour as well as estimates for (current and future) quantitative data points are many and thereby impractical/impossible to fully disclose. This places a large responsibility on the authors to be pedagogical in their writing, transparent in acknowledging and investigating uncertainties and their sources, and provide motivation for analytical choices. While the research presented in the paper

brings important contributions and insights on how to implement the sustainability transition proposed by the 6 transformation pathways, the manuscript needs to be revised and strengthened on these three points.

Major comments:

Please clarify shortly in the introduction how the transformative pathways are translated to the transition storylines. Most importantly, clarify how and when the elements called: barriers/impediments/bottlenecks, enablers/enabling conditions and accelerators/policy levers are determined. Without such clarification, it is difficult to read the results section – esp. the first half. Here, primarily, distinctions are needed on which (types of) elements are pre-determined (by previous research, the transformative pathways defined in previous literature, the relevant SSPs or the Australian context), which are chosen/assumed by the authors and which are somehow derived from the integrated modelling itself. The way the result section is written now, it is sometimes vague if these elements are inputs or outputs of the analysis. I would also encourage a streamlining of the wording (e.g. barriers/impediments/bottlenecks... if you mean the same thing it will be easier to read if you call it the same word).

Please also clarify how the transition storylines interacts with the iSDG model. You mention that "the study aligns with approaches that create a dialogue between models and socio-technical storylines": How is the research here presented iterative – going back and forth between storyline creation and iSDG model calibration? In this context, it would again be valuable to include a discussion or reflection on how much of the results come from pre-determined choices in the storylines? Who makes those choices (authors, 'experts', politicians, the majority..?) and how much do they constrain the model and the possible range of modelled results? How much different would the model results be if other (plausible) storylines were drawn? Even if these storylines are designed to maximize the transformation according to the 6 pathways, the paper would benefit from a critical discussion on how they, still, constrains the model?

The Method chapter, together with the extensive supplementary material, provides much of the details needed to decipher the approach as called for above, but the critical details mentioned above needs to be made more easily available to the reader. To summarize, the paper would therefore benefit from a concise paragraph of e.g. who writes the storylines , how these storylines constrain the model, and to separate out what parts of the pathways described are so to speak 'designed' (and thereby pre-determined) and what are results of the intricate system interactions.

Result section - connected to the above, it is not always clear what are results and what are comments or background information. Review and make sure to differentiate, for example, P. 5, L141-143 - is this a result from your work or more of a sidenote, based on the references noted? On P.14, L.391-392 - did you run the model with different sequences? E.i. is this a result or an acknowledgement of a prior/established knowledge. Also, the first sentence on p. 14 adds confusion - does your work really reveal "a broad range of measures" or are they predefined and you analyse how they will impact the system? (possible rephrase to "our results reveal HOW a broad range of measures [...] can/could accelerate...")

Please read through the Results chapter to check for similar unclarities.

Lastly, the analysis focuses on the national context and international drivers of change are exogenously defined - with sensitivity analysis performed. This is soundly done. Still, the past decade has shown that global developments may be more impactful on sustainable development than local plans and actions. Especially unexpected, disruptive events. A reflection on how much can reasonably be controlled with national levers would be beneficial to add.

On the discussion: While I would prefer a shorter discussion (see also article guidelines) with succinct insights that reemphasise the robustness of the results and their immediate applicability.

Minor comments:

P1. L27 – are we not already at or beyond the midpoint? And I would change “to the SDGs” to either “towards the SDGs” or “to meet the SDGs”.

P1. L28 – “...Summit in 2023 is for political guidance...” – remove for (or rephrase the beginning of the sentence)

P1. L34 – there are many relevant references to point to here. For example:

- Liu, J., Hull, V., Godfray, H.C.J. et al. Nexus approaches to global sustainable development. *Nat Sustain* 1, 466–476 (2018). <https://doi.org/10.1038/s41893-018-0135-8>

- Engström, R.E. et al. Succeeding at home and abroad: accounting for the international spillovers of cities’ SDG actions. *npj Urban Sustain* 1, 18 (2021). <https://doi.org/10.1038/s42949-020-00002-w>

- Bennich, T., Weitz N., Carlson, H. Deciphering the scientific literature on SDG interactions: A review and reading guide. *Science of The Total Environment* 728 (2020)

<https://doi.org/10.1016/j.scitotenv.2020.138405>.

P3. Figure 1 – In the upper left corner, I would expect there to be a quite significant ‘dip’ or downturn for the development towards SDGs (if that is what the “y-axis” is illustrating, even though the y-axis is not explicit) at the critical junction, and that also the BBS scenario needs to climb out of that dip? Is this just a visual choice to keep the curve flat? I would expect something similar to Suppl. Mat. Fig 1a-d to give a more realistic shape around the critical juncture?

P4 L.91 – The very best performing goals looks like SDG17. Why is it not mentioned in the text?

Please also explain briefly why SDG8 performs better in the BBS scenario compared to the SWT.

P4 L.108 – change to “the compounding crises described in the beginning of the results chapter” or similar. For clarity and readability.

P4. Figure 2b. Why is there only a sensitivity analysis presented for the SWT? Could there be a range of uncertainties around the BBS that (for some unexpected developments) make the BBS trajectory perform better - close to the “worst case” of the SWT. I suspect not, but without a sensitivity analysis around BBS it is difficult to know.

P4. Figure 2. could the resolution be sharper here? Esp. righthand side

P4. Please note somewhere (before presenting the results on the SDGs in e.g. Fig 2.) that there has been a selection of SDG targets and indicators. Now it is clear on p. 20 and in the supplementary material, but not before then.

P5. L115 – “sees”? Review choice of word.

P5. L131 (or the first time “transformation target” is mentioned) Please clarify that these focus on 2050, and in other ways how they relate to/differ from the SDG targets, or how they were defined (by whom etc).

Figure 3. Suggest writing out “...in year 2050” in the Chart title of 3b.

ALL FIGURES: Please be consistent with using small or large letters in the Figures and in the legends and in the manuscript text. (E.g. write Figure 3B if you want to keep capital letters (here “B”) in the figure, or vice versa stick with Figure 3b and then use “b” in figure, but don’t mix)

Figure 3. Write out an A (or a) by Fig 3a)/A

P.10 L.293. The discrepancy between the SWT and adding the impacts of all TPs in isolation is a very interesting and noteworthy result. However, I can not see this clearly in fig 5b. Is there any way this can be made clearer in the figure? Or clarified in the text how the reader can see this?

Figure 5c – check legend: BBB – what is this? BBS? Or TWS?

P13. L370 – check the sentence, seems to be a grammar error.

Throughout the text: Revise the internal references to the method chapter “(Methods)”. This is often not very useful, and not always needed (if a short description of the approach has been presented – see major revision). Where it can be helpful to look in the Methods chapter to get the details on some part of the results I propose you write a more detailed reference, to the sub-section within the Method chapter.

P 15 L461: I would propose finding a more novel reference to ensure this describes ‘to date’.

P.17 L.515 and 521 – make sure to be consistent in present/past tense – “develop” or “developed”.

P17. L540-544 – this sentence is too long.

Supplementary Material – could you format Figure 1 graphs so that the COVID effects mentioned in the main manuscript are visible (esp. 1.a) is difficult to spot a “v” trend...)

RESPONSE TO REVIEWER COMMENTS

Reviewer #1 (Remarks to the Author):

Modelling six sustainable development transformations and their accelerators, impediments, enablers, and interlinkages

This manuscript presents a national scenario analysis of the progress towards sustainable development goals (SDGs) in Australia, based on a simulation model called iSDG. The text is **well-written, and the figures and tables, summarizing an extensive analysis, are clear. I congratulate the authors for this systematic analysis.** However, I have three major concerns about this study:

Response: Thank you for this initial positive feedback on our draft manuscript. We have given due consideration to all the concerns and comments raised and have endeavoured address these in our revised manuscript, in particular the three major concerns identified below. This has greatly strengthened the manuscript in key areas and we thank the reviewer for the helpful feedback. Note that in response to comments from all reviewers, we have revised the terminology used from the 'Sustainable Wellbeing Transformation' or 'SWT' in the original paper to 'STP' or 'Six Transformations Pathway'. For each of the six main pathway elements, we now simply referred to these as 'transformations' (T1 to T6) rather than 'transformation pathways' or 'TPs'.

1) The manuscript lacks a clear objective and analytical focus. In Line 48-51, the authors say that they make a significant advancement to national modelling studies to address the prespecified research gaps, and focus on improving the understanding of accelerators and impediments. While acknowledging that the subject of SDG achievement is immensely broad and complex, I still find the objective statement unfocused and overly ambitious, also considering that the previous paragraph on the research gaps is not fully convincing either. Line 40 says that modelling studies have not so far included six transformations simultaneously, but there are many studies that take interactions of SDG transformations simultaneously, as the authors cite, therefore the reader wonders which specific SDGs or transformations are missing in those existing studies and what this study adds. Without such a clear objective, the extensive analysis presented in this manuscript seems dispersed.

Response: Thank you for raising this important point. We have revised our introduction to better clarify the key research gaps and the objectives and contribution of this study. The study was undertaken in the context of our broader research on the acceleration of transformations to achieve the SDGs by 2030, as well as long-term sustainable development by 2050. Our objectives relate not only to the advancement of SDG-related modelling, but also the study of transformations. It is important to note that while advancements have been made over the past five years, there are still many gaps in knowledge including on the nature of the transformations needed to achieve the goals and how they interact, what specific actions could accelerate progress, why progress has been so slow and what could be done to overcome impediments.

SDGs modelling is helping to gain insights in these areas, but there are many gaps in capabilities. These include not only the broad scope of the 17 SDGs which have limited coverage in many existing models, but also relate to improving transformation pathways approaches, providing findings that are actionable by policy makers and other actors, and consideration of social, political and behavioural impediments that have hampered progress on the SDGs to date. Modelling often 'glosses over' these impediments by assuming that policy makers take action or that technologies are

adopted. However, empirical research on sustainability transitions shows that systems are path dependent and several lock-ins result in systems inertia which prevents acceleration.

To address these comments, we have better articulated the gaps and objectives in the introduction section (before Figure 1) in track changes. This includes relocating some text on modelling gaps from the discussion section (which also addresses comments from Reviewer 3 to shorten the discussion section and remove subheadings). This has been assisted through feedback from other reviewers, including comments from Reviewer 2 which highlighted several main contributions of the paper: 1. providing a breakdown of (six) important individual transformations that contribute to the overall transformation pathway to the SDGs, 2. describing many of the interventions in each transformation with explicit policies (as opposed to reflecting them just via their desired outcome), and 3. linking qualitative transition storylines with quantitative modelling. Adding to this, we also advance research on multi-system interactions – exploring synergies and trade-offs between the six transformations as well as the SDGs. This has not been attempted to date to our knowledge.

As now outlined in the revised introduction, the study makes significant advancements on previous national modelling studies to explore how the six transformations that have been defined as imperative to SDGs achievement could be materialised in the Australian context and their medium- and long-term implications for sustainable development. We aim to advance knowledge on specific policies that can accelerate each transformation, their individual and combined impact on achieving the SDGs, and how the transformations interact with complex synergies and trade-offs. Using a transitions storylines approach, we also aim to systematically identify impediments to each transformation and enabling conditions to overcome these.

2) The manuscript does not employ a clear framework for model-based policy analysis, hence results in a confusion of terminology. For instance, from Figure 1 and the corresponding text, I understand that the authors use Transformation Pathway (TP) to refer to policy scenarios, and they analyze their impact on SDG-like targets, which are policy objectives. Then, there are 'accelerators' which are apparently also policy levers, and the figures show the impact of TPs on TPs (e.g. Figure 3 & 5). I suggest the authors to use a well-defined policy and scenario analysis framework, for instance similar to the XLRM framework developed at RAND Corporation or the good old system analysis framework described in Miser and Quade (1984) Handbook of Systems Analysis.

Response: Thank you for this helpful comment. We are familiar with the XLRM framework and have used it in our research. The main four elements are reflected in the research design for this study, however we recognise the benefit in making this clearer. We have now revised the terminology throughout the paper and revised Figure 1 to address these comments, and we include a more detailed and clearer description of the study design in the introduction directly preceding and following Figure 1. The terminology is now more clearly linked to the XLRM framework. Firstly, we change language around 'accelerators', 'investments', 'interventions' 'policy assumptions' etc. so that we consistently use only 'policies' or 'policy levers'. Relationships (R) are formulated in the iSDG-Australia 2.0 model which is described in the Methods and we now also include a new overview figure and description of the model in Supplementary Information (Supplementary Figure 8). We now also better acknowledge model limitations in the final paragraphs of the Discussion section (we also include a Supplementary Data File with model input data and projections, as well as 49 additional charts in Supplementary Figures 9-11 with the baseline model calibration results in response to other comments below). External factors (X) that are quantified in the model are used in our sensitivity analysis which in effect tests the robustness of the scenario interventions (also now reflected in Figure 1). These relate mainly to exogenous global drivers and vary between the BBS and SWT (now 'STP')

pathways. For clarity, we have now split these into a separate (and new) Supplementary Table 2. Outcome metrics (M) for 2030 are indeed the SDGs targets and indicators, however we also require a longer-term framework of targets for 2050 (as the SDGs end in 2030) for which we use ‘transformation targets’ – so there are two ‘Ms’. We can appreciate that this creates complexity and we have endeavoured to better explain this and guide the reader including through an improved Figure 1, introductory text and supporting tables.

However, it is important to note a few additional considerations that motivate the design of our framework presented in Figure 1. There have been many scenario modelling studies that have identified ambitious/robust/low regrets policy options to address sustainability challenges and yet in many cases governments have still failed to act. It is therefore also important to understand the impediments to decisive policy action and broader enabling conditions for such action if we are to move more quickly towards the SDGs. Consideration of these impediments and enabling conditions is often beyond scope for scenario modelling studies, and by including policy levers in scenario assumptions it is generally assumed that common impediments are overcome. Empirical research on sustainability transitions and transformations highlights that governments often fail to act due to different social, political and behavioural impediments. Similarly, governments can be compelled to act with the right enabling conditions, including support from coalitions and powerful actors, maturing technologies and innovations, pervasive narratives, new goals, crisis events, changes in government, etc. The fact that many studies gloss over these challenges has motivated us to also incorporate transition storylines which review these aspects. We therefore have additional factors – i.e. ‘impediments’ as well as additional ‘enabling conditions’ needed to overcome these impediments and support policy action. While these are difficult to quantify it is important that we begin to recognise these in the modelling literature if we want to effectively inform action. We have now better explained this addition to the study design and ensured that we use consistent terminology throughout the paper (i.e. also removing synonyms such as ‘barriers’, ‘bottlenecks’ etc.).

The choice of the six transformations to frame our diverse suite of policy levers is motivated by the recent SDGs science-policy literature which has promoted this framework for use by governments in national SDGs strategy formulation (e.g. the 2019 GSDR and Sachs et al 2019). It is also now being applied by a number of countries. By using this framework, we aim to build on this literature and ensure a link to SDGs policy and practice. The same framework will be used in the forthcoming 2023 GSDR to be presented to governments at the SDGs Summit in 2023 (and which we are involved in drafting). This is also important given the very broad scope of our study – i.e. if we simply list all of the policy ‘levers’ it would be a long and complex array of actions. By packaging these into the six transformations we are able to provide an intuitive and systemic structure that supports policy coherence while also studying their individual and aggregate effects. They are transformation pathway elements - combinations of interventions that support particular transformations needed to achieve the SDGs as identified by science.

Because of the interconnected nature of our systems model, each transformation will also potentially impact on other transformations – e.g. they could help to advance other transformations (synergies) or impede them (trade-offs). Our study explores these interactions as well – i.e. the ‘TP to TP’ effects in Figure 5 (NB: note also the change in terminology in this figure). In response to comments below, we have also improved these figures. Again, this is important in the SDGs context given the well-known interconnected nature of the goals. However, these multi-system interactions are poorly understood and are rarely addressed in modelling studies – another gap we seek to fill.

3) Transparency is a main issue both in the manuscript and the underlying analysis. The model and data used in this analysis is “available upon reasonable request”, which is not commonly seen

anymore in scientific studies. There is no description in the manuscript about the core components and assumptions of this model, and how they relate to the six transformations and their interlinkages analyzed in this study. A summary, at least, would have been useful. The model documentation cited in the bibliography tells us only the relationships, not how they are formulated. Furthermore, the validation and calibration procedure is described in the Methods, some performance metrics are presented in the supplementary material, yet there is no clear visualization of these validation results, whether the model replicates the historical data, at least. Considering that the manuscript presents this analysis as a decision support tool with almost point estimates, not for what-if analysis, the reliability of the model seems to be even more important. In the current manuscript, however, the reader is left with a black box.

Response: we thank the reviewer for this feedback and we have addressed this by including more information about the model (in Supplementary Figure 8 and supporting text) as well as 49 additional charts showing the calibration of the baseline projection against data for a broad range of economic, social and environmental variables in Supplementary Figures 9 (economic), 10 (social) and 11 (environmental). We have also included an additional Supplementary Data File as an appendix to the study which includes the model input dataset used for calibration as well as simulation results to 2050 for a broad range of indicators used to measure progress on the six transformations. For information, we also make the iSDG-Australia 2.0 model and input files and projections available to the reviewers in Vensim format for information (attached as a Zip folder with password for extraction = 'stpsdg2023'). We hope that these additional measures help to mitigate this issue. As the iSDG family of models is owned by the Millennium Institute the model is subject to third party restrictions and we are not able to make the model publicly available for download. We recognise that this is a limitation. However, the iSDG model is freely available for use by academics for research (non-commercial) purposes through a simple agreement between the Millennium Institute and a research institution. We have updated the Data Availability and Code Availability sections to reflect these changes.

It is also worth noting that several previous publications have used earlier versions of the iSDG-Australia model - including in Nature Sustainability (Allen et al 2019 – cited in the paper) and which have provided more details on the model as well as the baseline calibration. The model is a large system dynamics model with more than 30 modules and thousands of variables. As such, it is not feasible to provide detailed information on the formulas and model structure etc. in the manuscript, however we can appreciate that this results in less transparency regarding the structure and assumptions.

We have now also included an additional section on study and model limitations under the Discussion section. The focus of our study is on exploring how the six transformations needed to achieve the SDGs as identified by science could be materialised in the Australian context. We acknowledge that this does not consider a diverse array of scenarios and potential futures, however our previous research has undertaken broader scenario analysis with the model. The model outputs should be regarded as what-if projections and not predictions, as they are dependent on the policy and exogenous assumptions used in the model. There is of course the possibility that our assumptions are insufficient to capture the range of uncertainty in all model parameters. We have endeavoured to address this through our validation and sensitivity analysis procedures. We note that Reviewer 2 compliments 'the broad sensitivity analysis across multiple uncertainty dimensions' while Reviewer 3 notes that the sensitivity analysis is 'soundly done'.

Relatively minor comments:

- Line 81-83 : I doubt that the terminology of V-shaped and A-shaped dynamics is commonly used. Could you explain this further?

Response: We have now clarified this in the manuscript at the end of the first paragraph under 'Results'. The V-shaped terminology relates to the shape of the trend where this is a temporary fall (or jump) followed by a return to the trend. It is a well-known dynamic in the economics literature and commentary around the COVID pandemic – i.e. whether or not there would be a V-shaped recovery. However, the V-shape is much more evident in quarterly data rather than the annual data used in our model - For example, see: <https://www2.deloitte.com/au/en/blog/economics-blog/2021/australia-v-shaped-recovery.html>

- Figure 2 (and the other figures) presents the projections for aggregate SDG progress, which is the average percentage of progress towards each goal. While such aggregate metrics, such as the SDG index of SDSN, is commonly and unavoidably used, it bears the drawback of over-aggregation and losing the contextual information and insights. The motivation for choosing such aggregate metrics, and the limitations of it, is not discussed.

Response: We agree and acknowledge that such aggregation does lose the contextual richness of results for the full set of 80 indicators included in the analysis. It is also widely done in the literature for practical purposes given the very broad scope of the SDGs and the need to present key results and insights. We acknowledge that the averaging or aggregation of indicator performance at the goal level or for all goals glosses over contextual information on performance of specific indicators. However, it was necessary in our study for pragmatic reasons to present and discuss key study results for such a large set of indicators. We now include this in the Methods section under 'Targets for 2030 and 2050 and method for assessing progress'.

- In Figure 2 and others, sensitivity analysis is based on 10 uncertain parameters described in the SI. In such a large model, I find it hard to believe that there are only 10 specific uncertain parameters. This relates to the opacity of model structure and assumptions. Could you please elaborate on this?

Response: As we clarify in the study Methods under the section on 'Sensitivity analysis' we select ten exogenous parameters for the sensitivity analysis as recognised by the reviewer. We also mention that for very large models such as the iSDG, good practice is to focus on those parameters that are both highly uncertain and likely to be influential. This is what motivated our selection. We note that Reviewer 2 compliments 'the broad sensitivity analysis across multiple uncertainty dimensions' while Reviewer 3 notes that the sensitivity analysis is 'soundly done'.

- In Figure 3 and the corresponding discussion of the results, the authors refer to the social tipping concept and state that SWT trajectory results in an accelerated transition. Regarding the tipping concept, I recommend a more careful use of the term (<https://wires.onlinelibrary.wiley.com/doi/full/10.1002/wcc.813>). Regarding the results in Figure 3, I see that almost all transformation targets show an S-shaped trajectory also in the BBS case, and they have already demonstrated acceleration, maybe even passed the inflection point. SWT seems only to shift the saturation point up. Please elaborate and describe the results accordingly. Also, given that the reader does not know about the model structure, it is not clear if these S-shaped dynamics are created by endogenous model structure, or embedded in the model.

Response: The references to tipping points are mentioned in relation to acceleration dynamics, and specifically with regard to price-parity tipping points for technologies. While we appreciate that there may be some broader debate regarding 'social tipping points', such tipping points are a well-known feature of technology diffusion and there is an abundance of empirical research to back this up (e.g. Rogers' work on diffusion of innovations). We respond to the second issue relating to Figure 3 below the following comment.

- In Figure 3, projections start from 2016, which is quite in the past and data must be available for more recent years. Could you explain what motivates you not to start the model from a date closer to the present? This also is not consistent with other figures where projections start in 2020.

Response: We have now revised these charts to commence in 2020 in line with other figures and also in response to other comments raised by the reviewers. The projections in Figure 3 are simulated model outputs generated by the endogenous structure of the model. For each transformation, they represent the average progress made towards a set of targets for that transformation. These had originally commenced in 2016 as this is the starting year for the analysis of progress towards the SDGs targets as it is when the SDGs commenced. However, the trajectories for the two pathways don't diverge until the policy assumptions kick in and as such it makes sense to start them in 2020. We have also removed the interpolated lines in the charts in response to other comments received as these are somewhat contrived and give little added value. To avoid confusion, we have now adjusted the figures so that they all commence in 2020. Note that projections to 2050 for the indicators associated with each transformation are now included in the Supplementary Data File.

- The storylines underlying the transition pathways are described clearly in detail, yet not how they were developed. In the Methods, I read that the authors developed the storylines with desk research based on the relevant literature. In the SI, I see common dimensions included in each description. It would have been useful to develop those with a robust framework for literature review and storyline components, or to describe such a framework if it used. Additionally, Lines 527-539 are not describing the methods, but the content.

Response: Thank you for this suggestion. We have now included some additional information on the process used to develop the storylines. As outlined in the methods, the template used is consistent for each storyline and was informed by our broader research on transformation impediments and enabling conditions. In a separate review paper (Allen et al 2023 - cited), we undertook a comprehensive review of the literature on the acceleration of transformations (covering close to 200 studies). Through this, we identified common impediments and enabling conditions for transformation which we use to structure our storylines – in particular the techno-economic, socio-behavioural, and political-institutional impediments. In terms of the process, the templates were populated through literature review and synthesis. This literature was sourced through our review of academic literature on accelerating sustainability transformations, online queries (e.g. in Google Scholar) for academic and grey literature specifically pertaining to Australia including government roadmaps and strategies, existing scenario modelling and technical reports from thinktanks related to each of the six transformations, as well as the authors' own knowledge of relevant documents and research.

- I am skeptical about the methodological choice of summing up the fractional effects as synergies on the last column of Figure 5a, which is not meaningful metric. Please motivate this choice.

Response: We agree that this was not clear and we have changed the heading of the last column from 'synergies' to 'spillovers'. This value is meaningful in the sense that it shows the amount or percentage points of additional progress that each transformation makes towards the transformation targets of the other five transformations. It's true that these can be synergies or tradeoffs and that this should be a net effect. As such, we have relabelled this as 'spillovers' as we think this appropriately reflects these effects.

- Throughout the manuscript, the text is not very clear which arguments are based on the findings, and which are the interpretation and reflection of the authors based on the literature. For instance, Lines 405-411. I suggest a clearer distinction between model findings, TP narratives, and the broader discussion.

Response: We acknowledge that this distinction becomes blurred in places as we are trying to synthesise important insights from the entire study and also support this with the broader literature. To clarify, we have made several revisions to the manuscript, including the previous text in lines 405-11 so that the source of evidence for findings is now clearer. Similar revisions are made throughout the manuscript to provide clarity on the source of evidence – whether that be from our modelling results, or from the literature. In particular, we have substantively revised the Discussion section to ensure that this issue is addressed.

- In the SI, I suggest having Table 3 before Table 2, since the former has the narratives and the latter has the quantification of those.

Response: we have re-ordered these tables and adjusted the text to reflect the new ordering.

Reviewer #2 (Remarks to the Author):

Thanks for inviting me to review this interesting paper. The study uses a system-dynamics model to assess progress towards the SDGs in Australia, both for a trends-continued and an ambitious transformation scenario. I see **three important contributions in the paper**: i) linking qualitative storylines of different transformations with quantitative modelling, ii) providing a breakdown how different individual transformations contribute to the overall transformation pathway, and how they interact, iii) describing many of the interventions in the individual transformations with explicit policies (e.g. increasing education expenditure, or increased tax rates for higher incomes), as opposed to reflecting them just via their desired outcome.

However, I also have a number of questions and recommendations, some of them conceptual and some on modelling choices and results. If these can be addressed by the authors, the paper would certainly **make an important contribution to the literature on transformation pathways** towards the SDGs.

Response: Thank you for this initial positive feedback on our manuscript. We hope that we have provided adequate responses to the issues raised below.

Conceptual points:

1) I found the terminology sometimes a bit ambiguous and confusing: "Transformation pathway (TP)" seems to refer to multiple things:

- the scenario resulting from implementing a single transformation
- the set of interventions or levers (reflected through changing parameter settings in the model)

associated with a single transformation

- progress in a set of outcome indicators associated mainly with a single transformation, but when all transformations are applied (e.g. Fig. 3A)

Here it would help the reader if these related but different concepts could be separated more clearly. Besides that, recognizing that the transformations are not fully independent, I am not sure if TP is a good wording for single transformations – in the literature it seems to be used more commonly for referring to the overall transformation.

Response: thank you for this suggestion and for highlighting the confusion around the terminology. We had originally used the 'pathway' terminology for each transformation as they could be undertaken in isolation or combined. However, we agree that it is perhaps more important to use the 'pathway' terminology to reflect all six transformations combined. To address this as well as comments from other reviewers, we have now changed the name of the SWT scenario to Six Transformations Pathway (STP). Each of the six transformations is now simply referred to as a 'transformation' – or T1, T2, T3 etc. The targets used to measure progress on each transformation are still referred to as 'transformation targets' using the acronym 'TT' – or TT1, TT2, TT3... TT-ALL etc. These revisions are now reflected in the text and Figures – including Figure 3a which, for example, now shows the impacts of T1 on TT1, TT2, TT3 etc.

2) The authors have done a great job in documenting the narratives, modelling choices and indicators associated with each of the six transformation in the Supplement. However, for the reader of the main manuscript things are not always immediately clear. I would find it helpful to **briefly summarize the transformations (e.g. very short qualitative description, selection of key modelling choices & key indicators) in a short table in the main manuscript.**

Related to that, the discussion in the Results section focusses mainly on TP 2-4. Why was this choice made?

Response: thank you for this positive feedback. While we see the value of such a table, unfortunately due to the overall length of the paper and total word limit for the main text we are not able to include this table in the manuscript. Due to the large number of model settings and indicators associated with each transformation, any table would quickly become quite long. We acknowledge that this does reduce readability as readers are referred to the supplementary information where all of the details are provided.

In terms of the discussion in the results section, again this was a pragmatic decision due to the limited length allowed for the main text which meant that it was not possible to include all transformations. As such we selected the transformations 2, 3 and 4 as illustrative examples which included a larger share of the policies.

3) I am generally a bit skeptical about averaging together the breadth of the SDGs into single aggregate indicators of progress. In my view, such aggregate indicators lump together so many different things that important details can get lost. (Although I do acknowledge that this is widely done in metrics such as the SDG index).

However, my main concern related to such aggregate indices is that they can be prone to weighting biases, such as: i) the SDGs contain more human-development related goals and indicators than environmental ones, which makes it easier for high-income countries (high human development, but also high environmental pressures) to score high. ii) Models are incomplete in their coverage of indicators, and tend to cover some SDGs better than others. This potentially introduces a second weighting issue.

To overcome this, I would recommend two things:

a) in addition to the single aggregate metric, split it into a number of broad categories (could be just human development vs environment, or along the “five P” clusters of SDGs.)

b) make it more transparent how much individual SDGs contribute to the overall index, e.g. by adding the number of indicators for each SDG to Fig. 2A.

Response: thank you for this comment and we understand the concerns regarding indices which is also raised by Reviewer 1. As the reviewer notes, the approach is widely used in the literature including the SDG Index. We agree and acknowledge that such aggregation does lose the contextual richness of results for the full set of 52 targets and 80 SDGs indicators included in the analysis. However, it was necessary in our study for pragmatic reasons to present and discuss key study results for such a large set of indicators. We now reflect this in the Methods section under ‘Targets for 2030 and 2050 and method for assessing progress’.

In our study, the issues raised by the reviewer are ameliorated to some degree due to the method used for aggregation. As is good practice, we aggregate or average indicator values at the target level and then at the goal level. This ensures that each of the 17 goals contributes equally to the overall single index metric. In terms of balance across the economic, social and environmental dimensions, we have now reviewed the indicator set in Supplementary Table 5 and included an additional column to allocate each indicator to a dimension (see column D[#]). Summing these, we estimate that a breakdown of: Economic (E): 23 indicators; Social (S): 29 indicators; Environmental (V): 28 indicators. While not perfectly balanced, this gives an indication of comparable representation of the three dimensions in the indicator set. We have now reflected this in the Methods section under ‘Targets for 2030 and 2050 and method for assessing progress’.

4) The S-shaped acceleration curves hypothesis seems to rest largely on Figure 3. However, if I understand correctly, the “emergence phase” shown in panel A is not based on data but just an interpolation. Here I would strongly recommend to plot actual historical data, as I suspect the historic trends could look quite different from the suggested emergence phase. For some indicators, such as the environmental ones, it would perhaps not even show an emergence phase, but a worsening until a turnover point. [If it turns out to be impossible to obtain historic data for all required indicators, using the calibrated model for backcasting would be OK as a fallback]. In light of the updated Figure 3, it might be required to nuance the hypothesis of S-shaped acceleration.

Response: The reviewer is correct that the s-curve hypothesis is somewhat contrived given that the historical data is interpolated back to the model commencement year in 1990. In response to this comment as well as comments from Reviewer 1, we have now removed the historic period from the charts in Figure 3 and commenced the projections in 2020 to bring them in line with other charts in the manuscript. The new charts still display the acceleration period which is of most relevance for this study, and we have also removed the s-shaped references in the text in regard to these figures.

5a) I found the discussion of the interactions of transformations around Figure 5 a bit confusing. This is in part related to terminology (comment 1 above), but perhaps also the concepts behind the given numbers can be explained a bit more clearly. For example:

- Generally: I understand that many of the stated “%” numbers are changes/differences in the relative progress metric (which is measured in %)? Then I would recommend to use “percentage points” as a unit whenever referring to differences.

Response: Thank you for this helpful suggestion. In response to earlier comments as well as suggestions from other reviewers we have now revised the terminology used in the paper to improve clarity. In Figure 5, this now provides a distinction between the transformations (T) and the transformation targets (TT). We have also changed the units to percentage points when referring to the changes in relative progress.

- Related to “Synergies” column in Fig. 5A: I assume this is the row sum excluding the diagonal, correct? Is this straight sum really meaningful? Instead I would recommend to evaluate how transformation 1 improves indicators from transformations 2-6, and so on. Due to different numbers of indicators, this can differ from the straight sum.

- Also, the wording with “synergies” is a bit confusing in the caption – it can be misunderstood as the “Synergies” column only counting positive interactions which are previously labelled “synergies”.

Response: The reviewer is correct that the final column is the sum the previous columns (except the diagonals). It is meaningful in the sense that a higher value in this column reflects greater net (positive) ‘spillover’ effects from a single transformation on all of the other transformations. However, it is also true that it considers both synergies and trade-offs and as such the label ‘synergies’ is perhaps not clear. We have therefore relabelled this as ‘spillovers’ in the figure and reflected this change in the text.

- I suggest to add something like “all targets” to the header of Fig. 5B, to make the difference between panel A (transformation 1 on targets from individual transformations) and panel B (transformation 1 on all targets) immediately clear.

Response: correction made to Figure 5b.

5b) On “aggregation gains/losses” as in Fig. 5C and associated discussion. What you call “aggregation losses” is, I think, in part caused by the fact that for interacting transformations the order of “applying” transformations in the decomposition matters.

For example, in order to reduce emissions from electricity generation, one can make changes on the demand side (e.g. more efficient appliances), or on the supply side (decarbonize power generation). If the demand transformation is “applied first”, it has a fairly large effect on emissions (as every saved kWh is counted with a high emission factor). On the other hand, if the supply-side transformation is “applied first”, emission factors are already low so the demand transformation has a seemingly small effect on reducing emissions. This does, however, not mean, that the two transformations (supply / demand), involve areas of duplication (only in the sense that they both target emission reductions) or even trade-offs.

Now this is of course an artificial example as only in models we can do such a decomposition analysis. In the real world these two transformation would happen simultaneously – and here I would consider the demand transformation as synergistic with the supply transformation (lower demand facilitates decarbonizing supply).

Perhaps the quantification and discussion can be adjusted so that it gives a clearer picture of what are actual trade-offs?

Response: this is an insightful point and thank you for this suggestion. We have revised the text relating to Figure 5c so that the discussion is clearer and we unpack the main effects in more detail. In particular, material efficiency targets and additional taxes (T2) slow economic production, industrial output, and emissions, while at the same time measures to decarbonise the energy system (T4) both reduce material intensity, productivity and emissions while also stimulating growth in green

industrial output and material consumption. As pointed out by the reviewer, there may also be overlap in terms of the effects of demand and supply side measures. These result in complex feedbacks in the model which we can 'artificially' decompose but do not necessarily reflect trade-offs.

Modelling choices and results:

6) Australia-specific point: As noted in the narrative of transformation 4, Australia is currently the largest (or close second largest) exporter of coal worldwide. So a holistic SDG pathway for Australia has to include a phase-out of coal mining and exporting, at least in line with the global trend in phasing out these fuels, or perhaps even faster.

From the SI I gather that a 40% reduction in coal mining and 30% for oil and gas by 2050 is assumed in the transformation scenario, is that right? This is clearly not in line with the Paris agreement, for this a much faster phase-out would be required (e.g. for coal a phase-out by 2030 in OECD and around 2040 in non-OECD). The SSP paper by Riahi et al. is cited here as a reference – are these numbers taken from an SSP1 baseline scenario? If yes: a climate policy scenario (e.g. SSP1-RCP2.6) would have to be used instead, otherwise SDG 13 is not achieved.

Response: These % reduction figures were originally estimated based on the SSPs database. For our SWT (now STP) scenario we did indeed use the SSP1-2.6 pathway which we estimated as -40% below the SSP2 projection (-20%) for global coal demand which is a -60% reduction from 2020 levels. We have checked again the SSPs database and the SSP1-2.6 (IMAGE) pathway shows a reduction of -56.2% on 2020 levels by 2050. As such, this is comparable with these broad assumptions. However, we agree that this does not appear consistent with a net zero national scenario for Australia. More recently, the SSPs have included a SSP1-1.9 pathway which projects approximately -80% reduction in coal and oil by 2050 and an increase in gas of +14.7%. The IEA net zero scenario projects a decline in demand for coal (-90%), oil (-75%) and gas (-55%) by 2050. Oil production in Australia has been declining for years now, and one would expect thermal coal to be largely phased out by 2050, however gas is more complicated and depends on assumptions regarding CCS and BECCS. The IPCC 2022 analysis finds global modelled pathways that limit warming to 1.5C project gas consumption to decline with median values of -45% by 2050. We have now updated the global assumptions in Supplementary Table 2, using global values of coal and oil (-90%) and gas (-55%).

However, these assumptions are already reflected in the economic projections for the STP which estimate the effect of a phase out of coal and mining production and reduction in gas production on mining gross value added (GVA) in 2050. Based on the latest Australian data, the majority of mining GVA in FY2021 was from metal ores (~72%) while coal was 7.4% and oil and gas were 16% (most of which was gas). For the purposes of our study, we estimate a reduction in mining GVA of approximately -15% in 2050 compared to the reference level as a result of a phase out of coal and oil and reduction in gas. In the model, this is calibrated based on the pace of the green energy transition in Australia. However, Australia also has large reserves of critical minerals which are expected to be in high demand during the energy transition and as such the long-term impacts from phasing out of coal may be of limited economic significance. In terms of the global trade outlook for Australia, our sensitivity analysis also tests the sensitivity of projections to larger changes in the trade outlook for Australia.

7) Temperature change and climate impacts: Which SSP scenario is the ~1.8°C increase in GMT for the SWT scenario taken from?

Response: this is the projected global mean temperature increase in 2050 from SSP1-2.6 (IMAGE). We also use national climate projections for Australia from the CSIRO. Again, we test the impact of this assumption in the sensitivity analysis which more than covers the range of SSP projections (min/max values used are +1-3C by 2050).

Related: How are climate impacts considered in the model? I would expect climate impacts in Australia to amplify further from the current warming level to the ones assumed by 2050 (1.8°C – 2°C), so more wildfires, more droughts and rainfall extremes (even if national average precipitation stays constant as assumed).

I acknowledge that fully covering these impacts is challenging, but I would ask the authors to at least make a bit clearer to which extent they are covered, and where they see risks of the remaining impacts to jeopardize SDG progress.

Response: We agree with the reviewer that incorporating climate change impacts into modelling projections is a complex task. Many modelling studies fail to do so which is problematic. For the iSDG model, climate change impacts are introduced through the effects of an increase in global mean temperature on economic productivity, damages to infrastructure and loss of life due to increased risk of natural disasters, and effects of temperature change on biodiversity. The model also allows for exogenous adjustments to precipitation. Negative climate change impacts can also be ameliorated in the model through investment in adaptation. To address this comment and comments raised by other reviewers, we have now included a description of the iSDG model including how climate change impacts are introduced into the model in the Supplementary Figure 8 and summary information.

8) The result that carbon dioxide removal covers around 30% of the required emission reductions towards net-zero (Fig. 4) is a bit surprising. Given the known trade-offs of large-scale CDR it should be checked if this compatible with the rest of the SDG agenda. Is this all happening through afforestation (transformation 6), or also BECCS? What is the combined land area used for land-based CDR? Are side-effects on other SDGs (e.g. through land competition, water requirements, etc.) taken into account?

Also, related to point 7: Considering recent fires, and given amplifying climate impacts, I have some doubts if afforestation can really provide this amount of (long-term, stable) CDR in Australia. So perhaps the mitigation strategy needs to be adjusted, shifting more towards actual emission reductions on the energy side, and less reliance on CDR.

Response: Thank you for this comment. As we note in the manuscript and supplementary information, our net zero assumptions are based on the best available deep decarbonisation pathways research undertaken for Australia as modelled by ClimateWorks (2020). In our SWT scenario, we use ClimateWorks 1.5C All-in scenario as the basis for the necessary scale and pace of change, which is the most detailed bottom-up modelling available. This includes calibrating the model to deliver a comparable scale of GHG emissions reduction for different sectors as well as sequestration from afforestation. The model projections include a small contribution from CCS for reducing fugitive emissions as well as gas capture at waste disposal sites by 2050. The scenario also includes a large contribution from carbon forestry sequestration from approximately 8 Mha of plantings by 2030 (which is based on ClimateWorks). We agree with the reviewer that this is ambitious and uncertainties remain regarding implications of forest fires, however while bushfires release significant amounts of carbon dioxide, forests generally recover over time generating a significant carbon sink in the years following a fire (for example, see:

<https://www.dcceew.gov.au/climate-change/publications/estimating-greenhouse-gas-emissions-from-bushfires-in-australias-temperate-forests-focus-on-2019-20>).

In the iSDG model, reforestation is stimulated through SDG expenditure which increases the stock of forested land which is balanced with other stocks including agricultural land and is constrained by land availability. However, this is done at an aggregate level and as such the model does not consider potential local-level competition for land. This national aggregation remains a limitation of the model which we now acknowledge in the final paragraphs of the Discussion section where we address limitations.

9) Overall progress in the aggregate SDG achievement indicator slows down remarkably after 2030, only moving from 82% to 89% over 20 further years (compared to ~40% to 82% in the 2020-2030 decade). Why is this? Additional policies (or a strengthening of the already used policies) after 2030 could be considered.

Response: the reviewer is correct that progress after 2030 slows. This is partly because our SDG policy assumptions are only for the period up to and including 2030. We do this because this is the agreed timeframe for the SDGs and we are particularly interested in whether and how progress can be accelerated by 2030 and the longer-term effects of this. After 2030, our policy assumptions are focused on net zero achievement rather than the SDGs. While beyond scope for this study, it would be possible to include further SDGs investment beyond 2030. However, due to inherent trade-offs among the SDGs and challenges associated with achieving some targets (e.g. elimination), our previous research suggests that 100% SDG progress may not be achievable.

10) I compliment the authors for doing such a broad sensitivity analysis across multiple uncertainty dimensions. However, I would still hesitate to call the outcome “confidence bounds” and attach a probabilistic interpretation to them, for the following reasons: I understand that the variation of the input parameters reflects a range specified by the modeller, and not their true probability distribution (which is not known). Also, there are still uncertainties not considered (e.g. structural uncertainty related to the way dynamics are represented in the model). Perhaps “distribution of sensitivity runs” (or similar) described better what these bands represent. Related, on the results of the sensitivity analysis: It is interesting that the distribution of sensitivity runs extends almost exclusively to lower SDG achievements than the main SWT scenario. Can this be interpreted as saying that there is a substantial risk of much lower SDG achievement even if all the transformations of the SWT scenario take place? Or is it more a feature of how the sensitivity analysis and/or model was parameterized?

Response: Thank you for this comment and we are happy to adjust the terminology. The modelling software (Vensim) sensitivity analysis procedures uses the terminology ‘confidence bounds’ however we have adjusted this to ‘distribution of sensitivity runs’. In terms of the results from the sensitivity analysis, the reviewer is correct that the SWT (STP) scenario is at the higher end of the distribution. There are several likely reasons for this which are associated both with the SWT (STP) assumptions as well as the sensitivity ranges used. One key factor relates to assumptions relating to future climate change impacts - i.e. the SWT (STP) assumes that similar action to address climate change is undertaken globally which maintains an increase in global temperature at 1.8C by 2050. The sensitivity analysis min/max values are set from 1 to 3C increase by 2050 and simulations closer to the 3C increase likely contribute strongly to the bottom range distribution in SDG achievement. Similarly, data for Australia tends to be at the higher end of the sensitivity ranges used for future

governance assumptions and significant reductions in governance indicators would result in reduced performance in SDG achievement.

11) The large dip in poverty rates in the COVID years (Suppl. Fig. 4, 2a) is surprising, given that the change in inequality (2b) is much more moderate.

Response: the large dip in poverty in 2020 results from introducing a temporary cash payment transferred to households in 2020 which briefly shifted a percentage of the population above the national poverty line. While this resulted in a temporary improvement in income for those at the lower end of the income distribution, it had limited impact on overall income inequality.

Other points:

12) I would suggest to include “Australia” in the title to make it clear that it is a country-specific and not a global study.

Response: While we appreciate this suggestion from the reviewer, our preference is to maintain the current title. While the study is a national application with Australia as a use case, we feel the study will be of global relevance given the application of a global framework for understanding SDGs transformations and the increasing attention to acceleration of progress towards the SDGs in the lead up to the 2023 SDG Summit and publication of the 2023 UN Global Sustainable Development Report. We make it clear in the abstract that the study is focused on Australia.

Reviewer #3 (Remarks to the Author):

Reviewer comments to author:

The article presents a **timely and impressive research effort that takes on a challenging analysis of how six transformative pathways toward sustainability could be materialized in the Australian context**. It investigates how far these pathways – that in previous literature have been defined as imperative to SDG achievement – can push progress on the 17 SDGs by both the Agenda2030 ‘deadline’ of 2030 and by a more long-term vision of 2050.

Overall, the paper is well written and easy to read. The paper furthermore seems to be based on **robust and systematic qualitative and quantitative modelling**. However, to make this article acceptable for publication, the latter point here must be made clearer, and weaknesses of the approach acknowledged. This can partly be addressed in the method chapter, but primarily it needs to be concisely described in the introduction chapter and highlighted in the discussion.

The type of system dynamics modelling that is undertaken is appropriate for the research question and powerful for exploring system behaviour in transformation. However, to its nature, SD models also tends to become a ‘black box’ for anyone on the ‘outside’, where assumptions on (future) system behaviour as well as estimates for (current and future) quantitative data points are many and thereby impractical/impossible to fully disclose. This places a large responsibility on the authors to be pedagogical in their writing, **transparent in acknowledging and investigating uncertainties and their sources, and provide motivation for analytical choices**. While the research presented in the paper brings **important contributions and insights on how to implement the sustainability transition** proposed by the 6 transformation pathways, the manuscript needs to be revised and strengthened on **these three points**.

Response: Thank you for this positive and constructive feedback. We have endeavoured to address the reviewer’s concerns and feedback in the revised manuscript and respond to the various comments below. This includes through:

- *Discussion of model and study limitations at the end of the Discussion section*

- *A description of the iSDG model in the Supplementary Information (Supplementary Figure 8)*
- *Inclusion of additional Supplementary Figures (9-11) showing results of the model calibration for a large selection of 49 indicators.*
- *Inclusion of the model input dataset and a broad selection of simulation results as a supplementary appendix (Supplementary Data File)*
- *Provision of a copy of the iSDG-Australia 2.0 model in the Vensim software to the reviewers for information (attached as a Zip folder with password for extraction = 'stpsdg2023')*

Note that we have also now changed the name of the SWT scenario to Six Transformations Pathway (STP). Each of the six transformations is now simply referred to as a 'transformation' – or T1, T2, T3 etc. rather than TPs. This is in response to issues raised by other reviewers.

Major comments:

Please clarify shortly in the introduction how the transformative pathways are translated to the transition storylines. Most importantly, clarify how and when the elements called: barriers/impediments/bottlenecks, enablers/enabling conditions and accelerators/policy levers are determined. Without such clarification, it is difficult to read the results section – esp. the first half. Here, primarily, distinctions are needed on which (types of) elements are pre-determined (by previous research, the transformative pathways defined in previous literature, the relevant SSPs or the Australian context), which are chosen/assumed by the authors and which are somehow derived from the integrated modelling itself. The way the result section is written now, it is sometimes vague if these elements are inputs or outputs of the analysis. I would also encourage a streamlining of the wording (e.g. barriers/impediments/bottlenecks... if you mean the same thing it will be easier to read if you call it the same word).

Response: Thank you for this helpful comment and we acknowledge that our description of the study design and use of terminology required further work. We have now addressed this in the revised manuscript based on feedback from all three reviewers. Firstly, we have revised the terminology used throughout the manuscript and in Figure 1 so that it is clearer and better aligns with the XLRM framework – i.e. policy levers, exogenous factors, model relationships and performance measures (targets). We have also added additional paragraphs to the introduction directly before and after Figure 1 which explain the study design in more detail. Figure 1 has been revised to more clearly identify the key elements of the study design.

It is important to note that behaviour in the model is driven by the quantitative policy settings which are packaged for each transformation as well as the exogenous assumptions which vary between the BBS and SWT (now called STP). The projections for each individual transformation as well as all six transformations use the exogenous assumptions for the SWT(STP) scenario. All policy and exogenous assumptions are listed in the new Supplementary Table 2 and Supplementary Table 3 along with their sources. All the quantitative settings are derived or guided by the literature and existing research and datasets. The range of potential policy interventions is constrained somewhat by the model structure which has evolved over time as we have added more policies to increase functionality of the model, however there are still limits to the model capabilities – i.e. it can't model every conceivable intervention. In particular, the model does not include broader enabling conditions such as the formation of supportive coalitions, influence from powerful actors, or the broader political context and dynamics. These aspects are addressed in the qualitative transition storylines which are intended to complement the quantitative policy settings but are not direct inputs to the model.

Please also clarify how the transition storylines interacts with the iSDG model. You mention that "the

study aligns with approaches that create a dialogue between models and socio-technical storylines": How is the research here presented iterative – going back and forth between storyline creation and iSDG model calibration? In this context, it would again be valuable to include a discussion or reflection on how much of the results come from pre-determined choices in the storylines? Who makes those choices (authors, 'experts', politicians, the majority..?) and how much do they constrain the model and the possible range of modelled results? How much different would the model results be if other (plausible) storylines were drawn? Even if these storylines are designed to maximize the transformation according to the 6 pathways, the paper would benefit from a critical discussion on how they, still, constrain the model?

Response: thank you and our revised introduction and description of the study design now take this into account. In previous national modelling studies we have explored alternative plausible scenarios – (e.g. focused on economic growth, or green economy, or inclusive growth etc - see citation Allen et al 2019). These different 'what-if' scenarios do indeed result in different outcomes on the SDGs due to the different assumptions and settings introduced into the model. A sustainability-oriented scenario (nested within the global SSP1) is projected to make greater gains on the SDGs, but still falls well short of achieving the goals by 2030.

For this paper, we specifically aim to explore what might be possible in terms of achieving the SDGs if a more transformative approach is adopted, guided by the six transformations framework. The policy settings and storylines were developed by the study authors but draw heavily upon existing studies and literature and were designed to accelerate the six transformations in a way that maximises progress towards the SDGs by 2030. The transition storylines and the model policy settings are related and complementary – i.e. while the policy settings identify and quantify the specific policy actions that drive the behaviour in the model, the storylines explore what impediments will likely stand in the way of this ideal policy outcome and broader enabling conditions for these policies to be implemented. In this way we create a 'dialogue' because the policies would not likely be actionable or achievable unless the storylines play out. This of course does not mean they are the only potential future pathways for transformation, but they certainly help to identify common impediments and enabling conditions that are likely to be important for any pathway. These enabling conditions (e.g. supportive narratives, coalitions, powerful actors etc.) are important for transformations to succeed, yet they are often ignored in modelling studies as they are difficult to quantify. For our study, we consider these in a systematic way for the six transformations, however only in a qualitative sense.

The Method chapter, together with the extensive supplementary material, provides much of the details needed to decipher the approach as called for above, but the critical details mentioned above needs to be made more easily available to the reader. To summarize, the paper would therefore benefit from a concise paragraph of e.g. who writes the storylines , how these storylines constrain the model, and to separate out what parts of the pathways described are so to speak 'designed' (and thereby pre-determined) and what are results of the intricate system interactions.

Response: thank you for this suggestion. As per our responses above, we have now included a more detailed description of the study design in the introduction to the paper, before the revised Figure 1. We trust that these revisions address the issues raised. The storylines themselves don't constrain the model apart from guiding the policies that are then used as inputs to the model. Importantly, they provide an explanation of how the model projections resulting from the policy inputs might be enabled – beyond the strictly quantitative components.

Result section - connected to the above, it is not always clear what are results and what are

comments or background information. Review and make sure to differentiate, for example, P. 5, L141-143 - is this a result from your work or more of a sidenote, based on the references noted? On P.14, L.391-392 - did you run the model with different sequences? E.i. is this a result or an acknowledgement of a prior/established knowledge. Also, the first sentence on p. 14 adds confusion - does your work really reveal "a broad range of measures" or are they predefined and you analyse how they will impact the system? (possible rephrase to "our results reveal **HOW** a broad range of measures [...] can/could accelerate...")

Please read through the Results chapter to check for similar unclaritys.

Response: Thank you for these comments. We have now revised the text (previous L141-143, L391-2, and beginning of p.14) to clarify these issues. We have also carefully read through the manuscript to adjust other similar instances to improve clarity of the source – i.e model results or supporting literature. The intention here is to support our findings from the quantitative projections with the broader expert literature on transformations, however we can appreciate the need to clarify the source. In particular, the Discussion section has been substantively revised to address comments from all reviewers.

Lastly, the analysis focuses on the national context and international drivers of change are exogenously defined - with sensitivity analysis performed. This is soundly done. Still, the past decade has shown that global developments may be more impactful on sustainable development than local plans and actions. Especially unexpected, disruptive events. A reflection on how much can reasonably be controlled with national levers would be beneficial to add.

Response: We have now reflected on this issue in the final paragraphs of the Discussion section where we address study limitations. The reviewer is correct that a range of global drivers are exogenously defined and tested using sensitivity analysis. For the SWT (STP) we assume that other countries are also prioritising achievement of the SDGs and net zero. The sensitivity analysis suggests that some potential gains would be lost through worsening global conditions – such as higher average temperatures, changes in economic conditions and outlook (e.g. inflation, trade), or other global events that undermine governance in Australia. While these global conditions are influential, they do not minimise the need for local/national strategies and actions. Indeed, systems change will only come about through such local action which may indeed be triggered by global crises. There may of course be other extreme global events (e.g. global conflicts) which are not captured in our existing analysis and which we acknowledge in the study limitations.

On the discussion: While I would prefer a shorter discussion (see also article guidelines) with succinct insights that reemphasise the robustness of the results and their immediate applicability.

Response: Thank you for this suggestion. We have now revised and shortened the discussion section and removed the subheadings, as per the article guidelines.

Minor comments:

P1. L27 – are we not already at or beyond the midpoint? And I would change “to the SDGs” to either “towards the SDGs” or “to meet the SDGs”.

Response: text revised as suggested.

P1. L28 – “...Summit in 2023 is for political guidance...” – remove for (or rephrase the beginning of the sentence)

Response: text revised as suggested.

P1. L34 – there are many relevant references to point to here. For example:

- Liu, J., Hull, V., Godfray, H.C.J. et al. Nexus approaches to global sustainable development. *Nat Sustain* 1, 466–476 (2018). <https://doi.org/10.1038/s41893-018-0135-8>
- Engström, R.E. et al. Succeeding at home and abroad: accounting for the international spillovers of cities' SDG actions. *npj Urban Sustain* 1, 18 (2021). <https://doi.org/10.1038/s42949-020-00002-w>
- Bennich, T., Weitz N., Carlson, H. Deciphering the scientific literature on SDG interactions: A review and reading guide. *Science of The Total Environment* 728 (2020) <https://doi.org/10.1016/j.scitotenv.2020.138405>.

Response: the three suggested references are now included.

P3. Figure 1 – In the upper left corner, I would expect there to be a quite significant 'dip' or downturn for the development towards SDGs (if that is what the "y-axis" is illustrating, even though the y-axis is not explicit) at the critical junction, and that also the BBS scenario needs to climb out of that dip? Is this just a visual choice to keep the curve flat? I would expect something similar to Suppl. Mat. Fig 1a-d to give a more realistic shape around the critical juncture?

Response: Figure 1 has now been revised to incorporate this suggestion, as well as suggestions from other reviewers.

P.4 L.91 – The very best performing goals looks like SDG17. Why is it not mentioned in the text? Please also explain briefly why SDG8 performs better in the BBS scenario compared to the SWT.

Response: performance on SDG17 is based on three indicators associated with government revenue, deficit and interest on debt. Generally, Australia is projected to perform well on these indicators under both scenarios. For SDG8, the marginally higher performance under BBS is due to indicators relating to per capita GDP and real disposable income. This is partly due to the higher government revenue settings (tax) in STP. We have now clarified these results in the text.

P.4 L.108 – change to "the compounding crises described in the beginning of the results chapter" or similar. For clarity and readability.

Response: text revised as suggested.

P4. Figure 2b. Why is there only a sensitivity analysis presented for the SWT? Could there be a range of uncertainties around the BBS that (for some unexpected developments) make the BBS trajectory perform better - close to the "worst case" of the SWT. I suspect not, but without a sensitivity analysis around BBS it is difficult to know.

Response: sensitivity analysis is conducted on SWT (STP) as we desire to test the robustness of our policies to changing global exogenous drivers. We would expect the sensitivity of the BBS to be similar overall. We have previously run sensitivity analysis on the baseline iSDG-Australia model (see Allen et al 2019 cited in paper) which showed limited sensitivity of model projections. We have clarified this in Methods ('Sensitivity analysis').

P4. Figure 2. could the resolution be sharper here? Esp. righthand side

Response: we have now revised the right-hand figure based on feedback from other reviewers. The Figure 2(b) now ends in 2030. The resolution is now sharper.

P4. Please note somewhere (before presenting the results on the SDGs in e.g. Fig 2.) that there has been a selection of SDG targets and indicators. Now it is clear on p. 20 and in the supplementary material, but not before then.

Response: we have now made reference to the set of 52 targets and 80 indicators across all 17 goals in the Figure 2 description.

P5. L115 – “sees”? Review choice of word.

Response: correction made.

P5. L131 (or the first time “transformation target” is mentioned) Please clarify that these focus on 2050, and in other ways how they relate to/differ from the SDG targets, or how they were defined (by whom etc).

Response: revision made in the text to clarify that there are 67 transformation targets with values set for 2050 which are based on the larger set of SDGs indicators and considered of relevance for measuring progress towards each transformation.

Figure 3. Suggest writing out “...in year 2050” in the Chart title of 3b.

Response: correction made

ALL FIGURES: Please be consistent with using small or large letters in the Figures and in the legends and in the manuscript text. (E.g. write Figure 3B if you want to keep capital letters (here “B”) in the figure, or vice versa stick with Figure 3b and then use “b” in figure, but don’t mix)

Figure 3. Write out an A (or a)) by Fig 3a)/A

Response: corrections made to all figures using lower case letters.

P.10 L.293. The discrepancy between the SWT and adding the impacts of all TPs in isolation is a very interesting and noteworthy result. However, I can not see this clearly in fig 5b. Is there any way this can be made clearer in the figure? Or clarified in the text how the reader can see this?

Response: We have now clarified how the reader can calculate these values in the text – i.e. in Figure 5(b) by subtracting the results for the BBS (62%) from the results from the TPS (e.g. for TP4 it would be 70%-62%=8%). Similarly, for SWT: 92%-62%=30%.

Figure 5c – check legend: BBB – what is this? BBS? Or TWS?

Response: correction made to SWT.

P13. L370 – check the sentence, seems to be a grammar error.

Response: correction made.

Throughout the text: Revise the internal references to the method chapter “(Methods)”. This is often not very useful, and not always needed (if a short description of the approach has been presented – see major revision). Where it can be helpful to look in the Methods chapter to get the details on some part of the results I propose you write a more detailed reference, to the sub-section within the Method chapter.

Response: subheadings now added to Methods references.

P 15 L461: I would propose finding a more novel reference to ensure this describes ‘to date’.

Response: additional more recent reference added.

P.17 L.515 and 521 – make sure to be consistent in present/past tense – “develop” or “developed”.

Response: correction made.

P17. L540-544 – this sentence is too long.

Response: correction made.

Supplementary Material – could you format Figure 1 graphs so that the COVID effects mentioned in the main manuscript are visible (esp. 1.a) is difficult to spot a “v” trend...)

Response: Real GDP chart formatted to better show V shape, however given this is annual data the V-shape is not as apparent as the quarterly GDP data. This is because of the way that temporary lockdowns affected specific quarters of GDP results more than others.

Reviewers' comments:

Reviewer #2 (Remarks to the Author):

Thanks for the comprehensive response to the points raised in my first review. Most of my comments are addressed well. Together with the changes suggested by the other reviewers and implemented by the authors, this has improved the clarity of the manuscript substantially. However for a few of my points I'd like to follow up (see below). Before that, one point on the meta-level: For the second review round I would find a version of the manuscript with additions/changes highlighted in colour or track changes mode quite helpful, or alternatively the relevant changes in the manuscript being quoted in the response document together with the authors' response.

Follow up on my original comments:

#3: The added classification of indicators into economic, social and environmental indicators is indeed helpful. However, as already suggested in my original comment, it would be good to also include a quantitative breakdown of how the different dimensions contribute to the overall results. For a high-income country like Australia, one could imagine good performance for social and economic, but less good performance for environmental indicators, at least in early years. I would suggest to add three smaller panels with the results aggregated to the three dimensions economic, social, environmental to Fig. 2b. Or, if the authors prefer to not modify Fig 2b, a supplementary figure with this information could be added. (Although personally I would find it relevant enough for the main manuscript.)

#4: The removal of the interpolated S-curves removes a potential source of confusion. But I'm not so sure that the blanket removal of years prior to 2020 from the figure is the best solution. I would find showing some historical years (at least going back to 2010 or 2015) in the figure quite helpful, as this elucidates how the STP scenario breaks with historical trends, while the BBS should continue them more or less. Also from the response to a comment by R1 I understand that years prior to 2020 are modelled, so the authors could simply show these. Adding actual historical data on top would be even better (allowing for an evaluation of model performance in reproducing historical trends), but I acknowledge that this might be hard to come by for all indicators.

#12: Thanks for the explanation. Here I don't quite agree, for the following reasons:

- The SDG implementation challenge is very different for high-income and low-income countries (in terms of for which SDGs the gap is largest, how to finance the transformation, etc.).
- A country-level study can focus on achieving the SDGs in an individual country, whereas a study aiming to model global SDG achievement would also need to look into international collaboration required to move forward the SDG implementation (e.g. through financing investments into SDG-related infrastructure in the Global South with money from the Global North).

Along these two lines, I find the information that the study focuses on the national context in a high-income country so important that it should be reflected in the title (as the authors also did in their own 2019 paper in Nature Sustainability). At the same time, I don't want to overstep my role as reviewer, so would leave the decision to the authors and/or editors whether or not to change the title as suggested.

I was also asked by the editor to comment on whether the concerns raised by R1 are sufficiently addressed. While such an assessment by nature involves some second-guessing, and while some of the comments by R1 were at a more fundamental level than my own, I generally found that the authors responded thoroughly. Below, I comment on a few individual points raised by R1 and the associated responses.

- R1's point #2: As I haven't worked with the XLRM framework before, I can only assess this response superficially. But at this level, the mapping between the components of the analysis described in the paper and the building blocks of the XLRM framework make sense to me.

- R1's point #3: I generally agree with R1's point that transparency and reliability of models are an important issue. Concerning transparency, this means that the exact equations of the model should be available to the reader. While it is true that such a description cannot be included in each paper using the model, other integrated assessment modelling teams face the same issue, and have solved this by providing some combination of online documentation, documents detailing the equations, or even papers that describe a certain model version (detached from a particular scientific application). If such documentation is already available for the iSDG model, I would suggest to point to it more clearly. If not, I would encourage the authors to move into this direction with the iSDG model, too (while acknowledging that producing such documents is beyond the scope of this review).

- R1 also commented on the aggregation of SDG indicators and the problems associated with it. This echoes one of my own comments, so I would encourage the authors to implement the suggestion in my follow-up (#3 above).

Reviewer #3 (Remarks to the Author):

Comments to author:

This is a very interesting research paper, and with the revised manuscript I have much more confidence in the robustness of the analysis and the scientific advancements of the work. What still remains to be improved is significant clarifications, language checks and checking for repetition. While this may sound minor, it entails a need to restructure parts of the text, replace/redo/change figures etc. For this reason, I recommend major revisions of the manuscript before finding it appropriate for publication.

MAJOR COMMENTS:

Clarify the aim and objective of the study. This is national scale, which sets it apart from the work leading to the formulation of "the six transformations" (which was globally focused), but the nature/purpose of formulating the six transformations was that they should help to accelerate the SDGs. This basis makes it a bit strange to analyse how the SDGs progress if we follow them – a bit of an echo chamber. Rather, I would propose rephrasing the aim to "identify the type and magnitude of policy and investment needed to make these transitions really feasible" – in a national context, or similar. Then it makes sense to also "check" that the pathways really do meet/progress the SDGs, but that is to me not the main attraction here.

The reader needs to be too much of a detective in this paper. In particular, too much of the key parts of your research is placed in supplementary material. I can understand this from a space perspective, but I still urge you to move bits of this information into the main manuscript (I have never read a manuscript with so many references to the supplementary material). This includes, but is not limited to, a textbox/table detailing the storylines and their main references (but shorter than the Suppl. Table 3), some examples of TTs and possibly the Suppl. Table. 1.

Reversely, I believe you can reduce the text in the main manuscript that describes, previous research. Check for repetition on more general descriptions – such as that decisive policy action needs the right conditions. I would prefer more specifics on your analysis (that is now largely in the suppl. Mat. and less general comments – or at least be wary of their repetition).

Another element that needs to be described more clearly early on is that this is a 2-step analysis (right?). The method and the discussion is now more clear on this dual analysis (narratives/storyline development based on previous literature review and new detailed literature/source searches + SD modelling of these (quantified) storylines), but this clarity is missing from the introduction, which makes the results sections still a bit confusing. To give an example, result section on T3 start off by describing the current status/base year, followed by the key elements of the T3 scenario. In another paper, where the core of the analysis is the modelling, both of these paragraphs would have fitted better into a method or case description section (as would most of the first subsection of Results: "Covid-19 and cascading crises"). But if, in your study, these are really the results of a systematic search and identification (through literature mostly as I understand it, but please clarify also how this was done methodologically) of current conditions and plausible steps towards the (by others defined)

6 transformations, it may be the right place to keep these in the Results. Also, you mention that there is a "dialogue" between social science storylines/pathways and SD model(s) – is that in general/other's work or something you also do here? If the latter, please give some description of how this is done. If the main contribution of the paper is the SD modelling of these storylines/pathways, then I strongly suggest moving the descriptions of the scenarios out of the results section.

On that note, beware of your claims. It seems to me from the method chapter that several findings related to these storylines - and of what could act as impediments or enabling conditions... are findings already published in previous works by the authors or others. If this is true, you need to rephrase the wording in the results/discussion (?) to that you "confirm" these findings (from reference 33 etc.) or similar, rather than "We identify common impediments, [...] important actors, [...] enabling conditions [...]" (P16). Or, again, clarify if you in this work really do reveal the specific impediments, enablers etc. that have not been identified before?

//

To help the reader, there are also general logics of your narratives and modelling that would be valuable to state early/in the introduction. For example, as it reads on p. 14/15, it seems that "SDG policies" (guessing this entails regulation, subsidies, investments/expenditure) end in 2030, e.g. are drawn back/not continued, but it is a bit unclear. Are they replaced by new policies to push towards TTs – or are they still in place beyond 2030? I struggle to get this confirmed in the Supplementary material as it is written now, but more importantly, this type of general logic in the storylines and model projections needs to be stated in the main manuscript.

Another such general logic that is mentioned in the method, but would merit to be highlighted sooner in the text is that all T's are completely complementary (by design), and do not have any overlaps (e.g. no 2 T's have the same TT with different values)(right?). A sentence on this when/before you start discussing interactions would make that result section more engaging, so that it is clear that the interactions occur from indirect effects, and especially interesting to note the trade-offs then, and that the sum is indeed smaller than the parts, despite the T's being complementary (by design) (!).

In the Results chapter, I am still struggling to differentiate between model inputs and model outputs (see comment above on specifying that the inputs are also results –from the storyline creation process – if that is correctly interpreted). I find this to be, still, a critical issue, since the iSDG model is big and difficult to decipher as a reader, there has to be clarity between the modeller's design of the system and its development and what the model projects "by its own". This links back to the comment on "dialogue" between model and storylines. If it occurs in this work it needs to be described. If there is no dialogue, it needs to be made clearer what are inputs and what are outputs in the Results section (not only in the supplement mat.).

In the discussion I call for 2 things: Stick to discussing results already revealed in the Results section and draw new insights on what these results mean, how the different results relate, and how the results could have been different in different settings. Just to give two examples:

- what does it mean that there are synergies and tradeoffs, and how should e.g. T2 be valued/prioritized, given that it (a) contributes the least to the SDGs, but (b) enables many of the other T's (p. 8-9)?

- how can this be used, and by who?

The Discussion section is good, and many such good insights are highlighted, but more could be added. AND, importantly, I find new details on results that I find very interesting – but that should be raised in the Results section. For example, P15 lower half highlight some details that are new and important. If the developments described in L468 and L472 ("rapid transition to renewable electricity" and "additional investment in adaptation and resilience") correlate to specific TTs, then (1) write the TT number out!, and (2) I propose adding a larger and more detailed matrix (or figure) in the results chapter, that displays interactions/synergies on TT-level, with all TT's cross-interaction or similar. This would also show the interaction within the same 'T' that you mention in L480-481. There is a lot of work hidden but clearly done in this study, and the Manuscript would in my eyes benefit from displaying some additional figure or Table that more visibly portrays the level of details and scale of analysis. I believe you could remove one or 2 of the current figures and make space for one such more detailed matrix (see below!).

In the Method chapter, I would prefer a bigger focus on the how and less on the what. In my view,

several sections of the chapter outlines previous research and outlines the theory behind the analysis, more than the method. This is OK in smaller amounts, but please revisit and see if some of these theory/background paragraphs/sentences can go to the Introduction?

MINOR COMMENTS:

Please check for repetition between results/discussion and method/results.

Check old wordings (e.g. should "sustainable wellbeing framework" be changed on p.15, L455?) and check "TTs" where there should be only one "T" (see also commented PDF).

Is "wellbeing" the best title for T1? To me it seems to be more about resilience and social capability.

Don't use double-arrows from exogenous assumptions box in Fig 1. They are not affected by the 2 metanarratives – if they are, they would by definition not be exogenous.

You mention briefly XLRM above Fig 1, but don't describe it closer (and it seems to have the wrong reference). Meanwhile the X, L and M dimensions seems to be included in FIG 1. Could you also include R somehow, or is that not included in the analysis? (for example, synergies and trade-offs are system interactions rather than model relationships, but is R considered anywhere else?)

Beware of how you use "trajectory", "pathway", "scenario" and "projection". Please check that these word are used consistently and not overlapping. If you want/need to use all of these terms I would suggest a glossary at the beginning of the paper to clarify their differences.

When does the simulation start? By Fig 2b, it looks like the trajectories diverge before 2020. But in the text it reads that the critical juncture (from where to diverge from BBS) happens in 2022. Is the x-axis in Fig 2b wrong?

I suggest replace Fig 2b with Suppl fig 2. Now you refer to Suppl Fig 2 in the text but never to Fig 2b, and as far as I can see they are the same up until 2030 – then Suppl. Fig 2 provides additional info = is better to use. Even when you mention Fig 2b) on line 179, this note on sensitivity of results is also supported by Suppl. Fig 2. Then you can also remove Fig 3c.

Is Figure 3b providing any new information? Adding %numbers to Fig. 3a could make Fig 3 redundant (and even without %numbers, the relative gaps are clear, which is the point of Fig 3b as I see it.).

I don't see the value of Figure 4. It is briefly commented, but not in more detail than other results, and the detailed emission sectors are not at all mentioned in the text. I suggest remove (and add new more detailed figure/matrix as suggested above).

I doubt if Fig 5b is the best way to display what you want to highlight there (the 6% difference in all T's added up vs STP)? If you only discuss 2050, why show the timeline? Maybe just a table for 2050 would suffice, ending up in something that more clearly show that 6% difference?

I find Figure 5c still very confusing. I was trying to find a correlation between the column sum in Fig 5a and 5c, but find none (right?). But more troubling, the sum of all transparent bars in Fig 5c does not add to a net "trade-off" of 6%. Please review this and think if there is another way to explain it or display it. Fig 5a is in this context much clearer and better displayed.

Could Supplementary Table 1 be better placed in the main manuscript, potentially with also very short descriptions of T1-6. Or put only Table S.1 as it is in introduction (instead of referring to the Suppl. in parenthesis) and put a table summarising t1-t6 in methods, under Transition storylines. Here a column simply stating total # of TTs in each T would be useful (it is difficult to count/get a grip of this by rearing Suppl.Table.4 and would be good to just know if some T's have more TTs than others etc).

Supplementary Table 5: Could it be made easier to find the TTs for each T1-T6? Also, now there are a bunch of targets (e.g. all under SDG16 and 17) that has no T#, does that mean they are employed in all T's?

Detailed Response to Reviewers' Comments

Reviewer #2 (Remarks to the Author):

Thanks for **the comprehensive response** to the points raised in my first review. **Most of my comments are addressed well.** Together with the changes suggested by the other reviewers and implemented by the authors, this has improved the clarity of the manuscript substantially. However **for a few of my points I'd like to follow up** (see below). Before that, one point on the meta-level: **For the second review round I would find a version of the manuscript with additions/changes highlighted in colour or track changes mode quite helpful**, or alternatively the relevant changes in the manuscript being quoted in the response document together with the authors' response.

Response: Thank you for taking the time to read our revised manuscript and for confirming that most comments are now addressed well. We have now followed up on the remaining points and include additional changes in the manuscript in track changes mode. We trust that this now adequately addresses these final comments.

Follow up on my original comments:

#3: The added classification of indicators into economic, social and environmental indicators is indeed helpful. However, as already suggested in my original comment, it would be good to also include a quantitative breakdown of how the different dimensions contribute to the overall results. For a high-income country like Australia, one could imagine good performance for social and economic, but less good performance for environmental indicators, at least in early years. I would suggest to add three smaller panels with the results aggregated to the three dimensions economic, social, environmental to Fig. 2b. Or, if the authors prefer to not modify Fig 2b, **a supplementary figure with this information could be added.** (Although personally I would find it relevant enough for the main manuscript.)

Response: Thank you for this suggestion and we included a very similar figure in our 2019 study published in Nature Sustainability¹. As such, we did not consider it a novel addition for this paper, however we appreciate that this is important for clarity and provides useful additional information. We have therefore now included this as a new supplementary figure (Supplementary Figure 2) and included a reference to this in the Results (from Line 227).

#4: The removal of the interpolated S-curves removes a potential source of confusion. But I'm not so sure that the blanket removal of years prior to 2020 from the figure is the best solution. I would find **showing some historical years (at least going back to 2010 or 2015) in the figure** quite helpful, as this elucidates how the STP scenario breaks with historical trends, while the BBS should continue them more or less. Also from the response to a comment by R1 I understand that years prior to 2020 are modelled, so the authors could simply show these. **Adding actual historical data on top would be even better** (allowing for an evaluation of model performance in reproducing historical trends), but I acknowledge that this might be hard to come by for all indicators.

Response: Thank you for this comment and we have now revised these figures to commence in 2015. It is not technically feasible to go back earlier than this as the charts present projections of progress towards targets starting in 2015. We feel that some clarification is required regarding how these

¹ [1] Allen, C., et al., *Greater gains for Australia by tackling all SDGs but the final steps will be the most challenging*. Nature Sustainability, 2019. 2: p. 1041-1050 DOI: <https://doi.org/10.1038/s41893-019-0409-9>.

projections are produced in the model. The charts show projections of the % progress made towards targets using a 2015 baseline as described in methods (i.e. they are normalised from 0-100% progress starting with a baseline in 2015). As such, there are no historic values that can be included prior to 2015 as these would be zero. To address this comment, we have now updated the charts to commence in 2015.

#12: Thanks for the explanation. Here I don't quite agree, for the following reasons:

- The SDG implementation challenge is very different for high-income and low-income countries (in terms of for which SDGs the gap is largest, how to finance the transformation, etc.).

- A country-level study can focus on achieving the SDGs in an individual country, whereas a study aiming to model global SDG achievement would also need to look into international collaboration required to move forward the SDG implementation (e.g. through financing investments into SDG-related infrastructure in the Global South with money from the Global North).

Along these two lines, I find the information that the study focuses on the national context in a high-income country so important **that it should be reflected in the title** (as the authors also did in their own 2019 paper in Nature Sustainability). At the same time, I don't want to overstep my role as reviewer, so would leave the decision to the authors and/or editors whether or not to change the title as suggested.

Response: Thank you for this comment and we are happy to accommodate this. We have adjusted the title to include 'Australia', as suggested.

I was also asked by the editor to comment on whether the concerns raised by **R1** are sufficiently addressed. While such an assessment by nature involves some second-guessing, and while some of the **comments by R1** were at a more fundamental level than my own, **I generally found that the authors responded thoroughly**. Below, I comment on a few individual points raised by R1 and the associated responses.

Response: Thank you for this confirmation and your extra efforts in responding on behalf of R1.

- R1's point #2: As I haven't worked with the XLRM framework before, I can only assess this response superficially. But at this level, the mapping between the components of the analysis described in the paper and the building blocks of the XLRM framework **make sense to me**.

Response: Thank you for this confirmation.

- R1's point #3: I generally agree with R1's point that transparency and reliability of models are an important issue. Concerning transparency, this means that the exact equations of the model should be available to the reader. While it is true that such a description cannot be included in each paper using the model, other integrated assessment modelling teams face the same issue, and have solved this by providing some combination of online documentation, documents detailing the equations, or even papers that describe a certain model version (detached from a particular scientific application). **If such documentation is already available for the iSDG model, I would suggest to point to it more clearly**. If not, I would encourage the authors to move into this direction with the iSDG model, too (while acknowledging that producing such documents is beyond the scope of this review).

Response: Thank you for your suggestion and we apologise for not making this clearer. We have now more clearly directed readers by including an additional citation with a web link to the detailed model document for the iSDG model which is publicly available online. There is also a model website which

includes reports from other studies including additional information on the model formulations and applications, demonstration versions of the model, and journal publications that have used the model (including the iSDG-Australia model). We also provided a full executable version of the iSDG-Australia model to reviewers in our previous submission which includes all model formulations.

- R1 also commented on the aggregation of SDG indicators and the problems associated with it. This echoes one of my own comments, so I would encourage the authors to **implement the suggestion in my follow-up (#3 above)**.

Response: Thank you and we have now included the chart suggested above (Supplementary Figure 2).

Reviewer #3 (Remarks to the Author):

Comments to author:

This is a **very interesting research paper, and with the revised manuscript I have much more confidence in the robustness of the analysis and the scientific advancements** of the work. What still remains to be improved is significant **clarifications, language checks and checking for repetition**. While this may sound minor, it entails a need to restructure parts of the text, replace/redo/change figures etc. For this reason, I recommend major revisions of the manuscript before finding it appropriate for publication.

Response: Thank you for confirming the value of the research and your increased confidence in the results. And thank you for your many helpful suggestions and comments over the past two revisions which have helped to strengthen the clarity and robustness of the study in many areas. We appreciate your time and interest in our study, and we have done our best to address the additional comments below. We hope that the revised manuscript now adequately satisfies your concerns, in particular the final major comments as well as the more minor suggestions.

MAJOR COMMENTS:

Clarify the aim and objective of the study. This is national scale, which sets it apart from the work leading to the formulation of “the six transformations” (which was globally focused), but the nature/purpose of formulating **the six transformations** was that they should **help to accelerate the SDGs**. This basis makes it a bit strange to analyse how the SDGs progress if we follow them – a bit of an echo chamber. Rather, I would propose rephrasing the aim to “identify the type and magnitude of policy and investment needed to make these transitions really feasible” – in a national context, or similar. Then it makes sense to also “check” that the pathways really do meet/progress the SDGs, but that is to me not the main attraction here.

Response: We are sorry to hear that the aims of the study and the relevance of the six transformation framework are not yet clear. To address this comment, we provide further clarification in the manuscript. Firstly, from line 51 where we explain the increasing relevance of the six entry points framework for national governments. Secondly in the paragraph commencing line 64 where we further clarify the study aims, and the relevance and timeliness of the study including the suggestion from the reviewer.

The six transformation entry points used in the paper are promoted by global experts in the 2019 and now the 2023 UN Global Sustainable Development Report (GSDR). These latest 2023 GSDR is a key input into the UN SDGs Summit in New York this year and provides guidance for countries to help to

accelerate national progress towards the SDGs. The six entry points provide an organising framework for more coherent and transformative action by countries and have the best chance for accelerating progress towards the SDGs. While they stem from a global report, they are very much focused on supporting national implementation and we are very familiar with the framework as we are authors of the 2023 GSDR. Based on this framework, the transformations and the SDGs are intrinsically linked - the six transformations provide entry points for policy and the SDGs provide the targets for measuring overall performance and progress. Our study aims are along similar lines to those suggested by the reviewer, but focus more on providing new evidence on how the six transformations can be accelerated, what this would mean for the SDGs, and what key interlinkages and impediments also need to be considered. The value of this has been recognised in comments from R1 & R2. To make this clearer, we provide further clarification in the introduction as mentioned above.

The reader needs to be too much of a detective in this paper. In particular, too much of the key parts of your research is placed in supplementary material. I can understand this from a space perspective, but I still urge you to move bits of this information into the main manuscript (I have never read a manuscript with so many references to the supplementary material). This includes, but is not limited to, **a textbox/table detailing the storylines and their main references** (but shorter than the Suppl. Table 3), some examples of TTs and possibly the Suppl. Table. 1.

Response: Thank you for this comment and we acknowledge that there is a lot of supplementary information supporting the study as we are trying to ensure transparency with regard to all assumptions. In response to this issue (and within the space constraints), we have added an additional table (Table 1) which provides the two pathway metanarratives from the former Supplementary Table 1. We also add a brief synopsis of the storylines for T1 to T6 as Table 2 in the Methods. The breadth of the SDGs and the analysis undertaken involving all goals and 80 indicators unfortunately requires quite a lot of supporting material. Quite a lot of additional material also has been added in response to the peer review comments thus far. However, we appreciate the reviewer's concerns in this regard. We hope that this meets the needs for greater clarity while ensuring that the manuscript is as brief and streamlined as possible and acknowledging length limitations.

Reversely, I believe you can reduce the text in the main manuscript that describes, previous research. Check for repetition on more general descriptions – such as that decisive policy action needs the right conditions. I would prefer more specifics on your analysis (that is now largely in the suppl. Mat. and less general comments – or at least be wary of their repetition).

Response: thank you for this comment. We have now reviewed the manuscript for repetition and removed content relating to previous research, in particular in the Methods section. We have also now included more detail on the study methods in the introduction and Methods by providing a step-by-step overview of the process that was followed (please see also our response below).

Another element that needs to be described more clearly early on is that this is a 2-step analysis (right?). The method and the discussion is now more clear on this dual analysis (narratives/storyline development based on previous literature review and new detailed literature/source searches + SD modelling of these (quantified) storylines), but **this clarity is missing from the introduction**, which makes the results sections still a bit confusing. To give an example, result section on T3 start off by describing the current status/base year, followed by the key elements of the T3 scenario. **In another paper, where the core of the analysis is the modelling, both of these paragraphs would have fitted better into a method or case description section** (as would most of the first subsection of Results: “Covid-19 and cascading crises”). **But if, in your study, these are really the results of a systematic search and identification** (through literature mostly as I understand it, but **please clarify also how**

this was done methodologically) of current conditions and plausible steps towards the (by others defined) 6 transformations, **it may be the right place to keep these in the Results**. Also, you mention that there is a “dialogue” between social science storylines/pathways and SD model(s) – is that in general/other’s work or something you also do here? **If the latter, please give some description of how this is done. If the main contribution of the paper is the SD modelling** of these storylines/pathways, then I strongly suggest moving the descriptions of the scenarios out of the results section.

Response: thank you for this suggestion and we are sorry to hear that this clarity is still missing from the introduction. We fully acknowledge that this critical for the paper and we have now addressed this by revising Figure 1 and including a step-by-step overview of the process followed in the study (see from line 105 onwards). We hope that this now clarifies that the study involved several steps and also supports the logic of including both the findings from the modelling and the socio-technical analysis in the study results. An important aspect of the study approach is that it bridges modelling and socio-technical transition studies to provide a more comprehensive analysis of the transformation pathway. Using this approach, it is important to accompany the modelling results with findings from the socio-technical analysis and the transition storylines.

In brief, the study starts with the formulation of the metanarratives and exogenous assumptions for each pathway. Secondly, for each transformation (T) we reviewed empirical research, policies, technical reports and analyses, government data and other literature to inform policy choices and to provide a socio-technical analysis of current regime conditions, impediments, and promising niche innovations. The first two steps informed the development of quantitative policy settings for each pathway which serve as the key inputs to parameterise the model and project the pathways to 2050. In a fourth step, the quantitative projections for each transformation were confronted with the socio-technical analysis of key impediments. A future-oriented qualitative transition storyline was then constructed for each transformation which describes how impediments are overcome through particular socio-technical enabling mechanisms. The quantitative settings were then revisited to ensure consistency with the storylines, in particular with regard to the scale and speed of change. The performance of the final model projections for BBS and STP (and each individual transformation) were then benchmarked against the SDGs and transformation targets (TTs) (and we also now include a more detailed explanation of why we use these two sets of targets in the analysis).

The literature uses the word ‘dialogue’ to refer to this process, but we have now deleted this as it appears to generate some confusion. However, it is important and relevant for the study results to include both the model projections and our findings regarding the important impediments and enabling conditions for the policies included in the modelling to succeed.

On that note, beware of your claims. It seems to me from the method chapter that several findings related to these storylines - and of what could act as impediments or enabling conditions... are findings already published in previous works by the authors or others. If this is true, you need to rephrase the wording in the results/discussion (?) to that you “confirm” these findings (from reference 33 etc.) or similar, rather than “We identify common impediments, [...] important actors, [...] enabling conditions [...]” (P16). Or, again, clarify if you in this work really do reveal the specific impediments, enablers etc. that have not been identified before?

Response: thank you for this comment. We have now revised the wording in the results to make this clearer. We hope that the increased clarity regarding our methods and approach which we have outlined above also help to address these concerns. In particular, that the socio-technical analysis

and transition storylines used in the study are intended to inform and complement the modelling results with empirical findings from the literature on the more qualitative (e.g. socio-political) impediments and enabling mechanisms. These are not included in models and as such they cannot be identified through the modelling results. The intention is to enrich the modelling results with socio-technical insights from the empirical literature.

//

To help the reader, there are also general logics of your narratives and modelling that would be valuable to state early/in the introduction. For example, as it reads on p. 14/15, it seems that “SDG policies” (guessing this entails regulation, subsidies, investments/expenditure) end in 2030, e.g. are drawn back/not continued, but it is a bit unclear. Are they replaced by new policies to push towards TTs – or are they still in place beyond 2030? I struggle to get this confirmed in the Supplementary material as it is written now, but more importantly, this type of general logic in the storylines and model projections **needs to be stated in the main manuscript**.

Response: thank you for this comment and the reviewer is correct that the SDGs policies end in 2030 (as does the current SDGs framework). Beyond this we only include assumptions related to achieving net zero (which has a 2050 timeline). We have made this clearer in the introduction (in paragraph commencing 118) and it is also identifiable in Figure 1. The SDGs policy period is also highlighted using shading in Figure 3.

Another such general logic that is mentioned in the method, but would merit to be highlighted sooner in the text is that all T’s are completely complementary (by design), and do not have any overlaps (e.g. no 2 T’s have the same TT with different values)(right?). **A sentence on this when/before you start discussing interactions** would make that result section more engaging, so that it is clear that the interactions occur from indirect effects, and especially interesting to note the trade-offs then, and that the sum is indeed smaller than the parts, despite the T’s being complementary (by design) (!).

Response: thank you for this suggestion and the reviewer is correct that each T has a unique set of TTs. We acknowledge that the evaluation framework used in the study has created some confusion, and we now more clearly explain this in the introduction (paragraph commencing line 134) and methods.

To evaluate progress on the SDGs we use a unique set of 80 SDG indicators with target values for 2030 and 2050. A subset of 67 of these indicators are also used to evaluate progress made on the six transformations by 2050, which we label as ‘TTs’. Each T is allocated a unique set of TTs based on thematic relevance (e.g. energy-related SDG indicators are used as TTs for T4 on energy access and decarbonisation). We avoid duplication of TTs across the different transformations and exclude means of implementation indicators (from SDGs 16 and 17) which could apply to all transformations. While there is considerable overlap between the 2050 SDGs targets and the TTs, they assist with the analysis as they are aggregated differently (i.e. at the goal level for the 17 SDGs or at the transformation level for the six Ts). This provides several advantages - It enables us to compare the performance of the two pathways on all 17 SDGs in 2030 and 2050, to decompose the comparative impact of each individual transformation on the SDGs, to understand the comparative progress made on enabling each transformation and acceleration dynamics resulting from different packages of policy levers, and finally to reveal spillover effects from one transformation to another.

It’s also important to note that the system dynamics modelling is exploratory in that it explores the effects of each package of policies (grouped by transformation) on the SDGs and TTs. In this regard, a set of (e.g. energy) policies designed to achieve a set of TTs (e.g. for energy decarbonisation) can also

have synergies and trade-offs for other transformations and their TTs (e.g. for the environmental commons or urban systems etc.).

In the Results chapter, I am still struggling to differentiate between model inputs and model outputs (see comment above on specifying that the inputs are also results –from the storyline creation process – if that is correctly interpreted). I find this to be, still, **a critical issue**, since the iSDG model is big and difficult to decipher as a reader, there has to be clarity between the modeller’s design of the system and its development and what the model projects “by its own”. This links back to the comment on “dialogue” between model and storylines. **If it occurs in this work it needs to be described**. If there is no dialogue, it needs to be made clearer what are inputs and what are outputs in the Results section (not only in the supplement mat.).

Response: thank you for this comment and we understand that this also refers to the previous comment above regarding the clarity of the study methods. We have responded to this above and here by more clearly describing the step-by-step process followed in the introduction (from paragraph commencing line 105). We have also revised the Results for T2, T3 and T4 (from line 297) so that it is very clear which results relate to the sociotechnical analysis, which are from the modelling, and which specific policies are included in the model. It may also be helpful to note that all of the quantitative model inputs used for parameterisation of policies are documented in Supplementary Table 4 along with the sources for these settings.

In the discussion I call for 2 things: Stick to discussing results already revealed in the Results section and draw new insights on what these results mean, how the different results relate, and how the results could have been different in different settings. Just to give two examples:

- what does it mean that there are **synergies and tradeoffs**, and how should e.g. T2 be valued/prioritized, given that it (a) contributes the least to the SDGs, but (b) enables many of the other T’s (p. 8-9)?
- **how can this be used**, and by who?

The Discussion section is good, and many such good insights are highlighted, but more could be added. AND, importantly, I find new details on results that I find very interesting – but that should be raised in the Results section. For example, P15 lower half highlight some details that are new and important. If the developments described in L468 and L472 (**“rapid transition to renewable electricity”** and **“additional investment in adaptation and resilience”**) correlate to specific TTs, then (1) write the TT number out!, and (?)

Response: Thank you for this comment. We understand that the reviewer finds that the discussion section has many good insights, but would like to ensure that the discussion focuses on results already presented in the results section. We have now cross-referenced between the results and discussion sections to ensure that these sections are consistent. With regard to the specific issues raised, we would like to provide the following additional information:

- *We agree with the reviewer that interlinkages (synergies and trade-offs) are indeed an important feature for the discussion section. These are discussed in three paragraphs (from line 546). We have added some additional text for clarity.*
- *We also some additional text to indicate how and by who this information can be used.*
- *With regard to T2 – we have better clarified this in the results – i.e. the tax reforms in T2 are of critical importance for all Ts as they provide financing for policy interventions for all transformations. This should be prioritised because without this funding other policy reforms would not be achievable. Another key mechanism for doing this (a wellbeing budget) also forms part of the T2 storyline.*

- *With regard to the ‘rapid transition to renewable electricity’ and ‘additional investment in adaptation and resilience, these are linked in the text to the relevant transformation (T4 and T1, respectively).*

(2) I propose adding a larger and more detailed matrix (or figure) in the results chapter, that displays interactions/synergies on TT-level, with all TT’s cross-interaction or similar. This would also show the interaction within the same ‘T’ that you mention in L480-481. There is a lot of work hidden but clearly done in this study, and the Manuscript would in my eyes benefit from displaying some additional figure or Table that more visibly portrays the level of details and scale of analysis. I believe you could remove one or 2 of the current figures and make space for one such more detailed matrix (see below!).

Response: thank you for this suggestion. While we acknowledge that this proposal from the reviewer sounds like an interesting analysis, it is not technically feasible in the current study design as we do not assess interactions between the TTs. This is not how we modelled the interactions. Our approach is exploratory in that it implements a package of policies (for each of the six transformations) which then has an impact on the targets. We can explore the impacts of the transformations on the SDGs targets and on the TTs, but we cannot present interactions from one target (e.g. TT) to another target (e.g. TT). As such, it is not possible to produce such a chart or matrix in the current study, but we recognise that this could be an area for future for further work. The interactions that we are capable of presenting (e.g. between transformations and the SDGs, and transformations and the TTs) are currently included in Figures 5 and 6, and we have modified Figure 5 to improve readability. We hope that this clarifies this issue.

In the Method chapter, I would prefer a bigger focus on the how and less on the what. In my view, several sections of the chapter outlines previous research and outlines the theory behind the analysis, more than the method. This is OK in smaller amounts, but please revisit and see if some of these theory/background paragraphs/sentences can go to the Introduction?

Response: thank you for this suggestion. As per our response above, we have reviewed the Methods and other sections and removed descriptive text, as suggested by the reviewer. We have also provided greater clarity on the methods in response to the issues raised above.

MINOR COMMENTS:

Please check for repetition between results/discussion and method/results.

Check old wordings (e.g. should “sustainable wellbeing framework” be changed on p.15, L455?) and check “TTs” where there should be only one “T” (see also commented PDF).

Response: thank you. We have checked these references to sustainable wellbeing framework. We are not referring here to the previous pathway name included in the study method, but rather making a recommendation. As such, this seems appropriate to us. All wordings for T and TT have also been checked.

Is “wellbeing” the best title for T1? To me it seems to be more about resilience and social capability.

Response: it is true that T1 includes resilience, but it also includes other aspects of human wellbeing (e.g. education, health). Please note that we use the same titles for the Ts as used in the UN GSDR for consistency with this literature. We have made it more clear in Table 2 what is included in T1.

Don’t use double-arrows from exogenous assumptions box in Fig 1. They are not affected by the 2 metanarratives – if they are, they would by definition not be exogenous.

Response: we have removed the double arrows from Figure 1.

You mention briefly XLRM above Fig 1, but don't describe it closer (and it seems to have the wrong reference). Meanwhile the X, L and M dimensions seems to be included in FIG 1. Could you also include R somehow, or is that not included in the analysis? (for example, synergies and trade-offs are system interactions rather than model relationships, but is R considered anywhere else?)

Response: thank you and we have now clarified that the model relationships are formulated in the system dynamics model used in the study. We have made this clearer in the text above Figure 1 and in the Figure 1 explanatory text. Note that inclusion of the reference to the XLRM framework was done at the request of R1.

Beware of how you use "trajectory", "pathway", "scenario" and "projection". Please check that these word are used consistently and not overlapping. If you want/need to use all of these terms I would suggest a glossary at the beginning of the paper to clarify their differences.

Response: thank you and we have reviewed the text for consistency and removed the use of 'trajectory' and 'scenario' in reference to our pathways to avoid confusion.

When does the simulation start? By Fig 2b, it looks like the trajectories diverge before 2020. But in the text it reads that the critical juncture (from where to diverge from BBS) happens in 2022. Is the x-axis in Fig 2b wrong?

Response: thank you for highlighting this. The simulation commences in 1990, while the assessment of progress towards targets commences from 2015 and the alternative policies commence in 2021. We have revised the text to indicate that the pathways diverge in 2021, and we have revised and corrected Figure 2 to commence in 2015 at the request of R2.

I suggest replace Fig 2b with Suppl fig 2. Now you refer to Suppl Fig 2 in the text but never to Fig 2b, and as far as I can see they are the same up until 2030 – then Suppl. Fig 2 provides additional info = is better to use. Even when you mention Fig 2b) on line 179, this note on sensitivity of results is also supported by Suppl. Fig 2. Then you can also remove Fig 3c.

Response: thank you. We have now replaced the figure as requested and included additional information in the introduction to explain the target frameworks used in the study. Given that we use two frameworks for 2050 for different purposes – the SDGs targets and the TTs – we have retained the former figure 3c but removed 3b as per below.

Is Figure 3b providing any new information? Adding %numbers to Fig. 3a could make Fig 3 redundant (and even without %numbers, the relative gaps are clear, which is the point of Fig 3b as I see it.).

Response: thank you. We have removed former Figure 3b and included %s in 3a.

I don't see the value of Figure 4. It is briefly commented, but not in more detail than other results, and the detailed emission sectors are not at all mentioned in the text. I suggest remove (and add new more detailed figure/matrix as suggested above).

Response: thank you for this suggestion and we have now included some additional text to better incorporate Figure 4 into the study. We feel it is important to show that the STP also delivers a net zero outcome with some breakdown of key sectors as it is important for longer-term coherence and we better reflect this in the text. We think it will be of interest to readers given the importance of achieving net zero over the long-term by 2050. Please also see our response above to the previous

suggestion for including an additional matrix. As this is not possible to include such a matrix, we would prefer to keep the current Figure 4 along with some additional text.

I doubt if Fig 5b is the best way to display what you want to highlight there (the 6% difference in all T's added up vs STP)? If you only discuss 2050, why show the timeline? Maybe just a table for 2050 would suffice, ending up in something that more clearly show that 6% difference?

Response: thank you and we have replaced this figure with a simplified version in 5(b) which presents the difference in 2050. We trust that this improves clarity.

I find Figure 5c still very confusing. I was trying to find a correlation between the column sum in Fig 5a and 5c, but find none (right?). But more troubling, the sum of all transparent bars in Fig 5c does not add to a net "trade-off" of 6%. Please review this and think if there is another way to explain it or display it. Fig 5a is in this context much clearer and better displayed.

Response: thank you for this suggestion regarding Figure 5(c). We acknowledge that this was a complex chart to interpret and we have now revised all of the figures in 5(a) to (c) to display the calculations (in 5a) and to improve clarity as well as clearer descriptions in the text. Note that the average performance of STP on all TTs is calculated as the AVERAGE(TT1:TT6). This means that the 6% discrepancy is the average of the aggregation gains/losses and therefore isn't comparable to the sum of these values. We have revised Figure 5c to more clearly show this and also revised the text.

Could Supplementary Table 1 be better placed in the main manuscript, potentially with also very short descriptions of T1-6. Or put only Table S.1 as it is in introduction (instead of referring to the Suppl. in parenthesis) and put a table summarising t1-t6 in methods, under Transition storylines. Here a column simply stating total # of TTs in each T would be useful (it is difficult to count/get a grip of this by rearing Suppl.Table.4 and would be good to just know if some T's have more TTs than others etc).

Response: thank you for this suggestion and this also relates to the previous comment above. In response to both, we have now included the metanarratives (Supplementary Table 1) in the introduction and a table (Table 1) with a brief description of the storylines in the Methods (Table 2).

Supplementary Table 5: Could it be made easier to find the TTs for each T1-T6? Also, now there are a bunch of targets (e.g. all under SDG16 and 17) that has no T#, does that mean they are employed in all T's?

Response: thank you for these questions and we have included some additional explanation of the targets in the introduction (paragraph commencing line 145). The TTs are a subset of 67 indicators from the larger set of 80 SDG indicators included in the model. Each transformation is allocated a unique set of TTs based on thematic relevance (e.g. energy-related SDG indicators are used as TTs for T4 on energy access and decarbonisation). We avoid duplication of TTs across the different transformations and exclude means of implementation indicators (from SDGs 16 and 17) which could apply to all transformations. Including these as TTs for all Ts would have made interpretation of the results less clear. In the UN GSDR framework that we use in the study, these goals are considered 'levers' which are to be implemented for all six transformations. Supplementary Table 5 includes all of the targets used in the study and groups these by SDG and identifies their link to the transformations, however we can appreciate that it takes time to read through this. To improve clarity, we have also now included an additional table with the storylines under Methods (Table 2) which includes the number of TTs relating to each T and a brief indication of thematic topics.

REVIEWER COMMENTS

Reviewer #2 (Remarks to the Author):

I am happy to confirm that my follow-up comments are addressed appropriately. Thanks for putting together this impressive piece of research, I appreciated the opportunity to review it.

Reviewer #4 (Remarks to the Author):

Dear Authors:

I was asked to evaluate this manuscript for its ability to respond to R3 in particular. In general, I find this work to be interesting, compelling and important. But I think it would really benefit from taking some of R3's comments a bit more to heart. For example, the overall purpose of the manuscripts is a bit confusing. Take the abstract, which argues that there is a need to accelerate SDG development and that six pathways have been identified in new research. This research uses an IAM to identify how these pathways could be unlocked. And then the finding is that crises create the space for pathways to be pursued (?) and that this leads to good outcomes.

There are a few chickens and a few eggs. First, there are pathways that have been identified that allow us to understand how to pursue the SDGs. That comes first. So, is this manuscript really identifying *how* to pursue these pathways? I don't get that message from the manuscript. Is the manuscript telling me that a crisis space has facilitated the achievement of these results? That's not really covered in the manuscript either, and certainly isn't the main point.

If the point is to say that the SDGs can be achieved if you pursue these practical policies, then you need to focus on the specific interventions that get you to the scenarios you're building. How do we know that these policies are easily achieved? What is the magnitude of the interventions we're pursuing and how do we know that's a reasonable target? The paper is now about target setting and policies in relation to Australia's pursuit of these global pathways.

Is this a paper about synergies and trade-offs? If so, that should be the focus. You talk about the energy system as being essential and highly synergistic. That's a potentially useful finding, but then the abstract is focused on that.

I think R3 also has good points about the framing being more focused on a country level application of a global development pathway framework. There is less in this piece than I would expect about how a global framework could be applied to a national framework. And this global framework is grey literature (I believe), so that is an additional challenge. It's an especially important omission because the SDGs in high income countries are extremely different than LDCs both in narrative and empirics.

I think there is a tremendous amount of "meat" to this analysis and it should be published. But you need to clarify the purpose of the analysis earlier, more clearly, and make broader changes to reflect these concerns.

Minor comments relate to redundancies in the text (R3 is right about this, I think) and Figure 3: the line graphs show a spike or dip in the year the scenarios take that doesn't make sense. Why does the economy fall in the BBS scenario so quickly? Why the spike in food availability?

Response to Review

Reviewer #2 (Remarks to the Author):

I am happy to confirm that my follow-up comments are addressed appropriately. Thanks for putting together this impressive piece of research, I appreciated the opportunity to review it

Response: we would like to express our gratitude to the reviewer for their time and constructive feedback on our manuscript which has strengthened many aspects of the paper. Thank you!

Reviewer #4 (Remarks to the Author)

Dear Authors:

I was asked to evaluate this manuscript for its ability to respond to R3 in particular. In general, I find this work to be interesting, compelling and important. But I think it would really benefit from taking some of R3's comments a bit more to heart. For example, the overall purpose of the manuscripts is a bit confusing. Take the abstract, which argues that there is a need to accelerate SDG development and that six pathways have been identified in new research. This research uses an IAM to identify how these pathways could be unlocked. And then the finding is that crises create the space for pathways to be pursued (?) and that this leads to good outcomes.

Response: thank you for taking the time to review our manuscript. Your comments are much appreciated and have helped us to fine tune the framing of the study and ensure that the purpose of our analysis and the key findings are clearer.

We acknowledge that the scope of the analysis is quite broad – encompassing all six transformations and all SDGs, and that we aim to address several important research questions relating to the implementation of the SDGs. We also acknowledge that the methods are quite novel, in that we bridge the systems modelling with socio-technical analysis and transformation storylines in order to provide a more comprehensive analysis of the transformation pathways.

We have given a lot of thought to the comments from the reviewer, and we appreciate the need for greater clarity regarding the purpose and approach. In response to this comment and the other comments below, we have made multiple revisions (in track changes) to the abstract, introduction and discussion sections to improve clarity and to elevate the key policy findings. Specifically with regard to the study purpose, please see revised abstract (from line 15) and introduction (from line 69 onwards) where we define this more clearly. We more clearly specify the aims as follows:

“We aim to advance knowledge in four critical areas: (i) specific policy packages that can accelerate each transformation, (ii) the individual and combined impact of these policies on achieving the SDGs and net zero targets, (iii) the complex interactions between the transformations and the SDGs and their synergies and trade-offs, and (iv) important impediments and conditions that determine the success of policy reforms.”

Also note that our study uses a system dynamics model which is a bit different to an IAM, however the reviewer is correct that we aim to show how the transformations can be unlocked and accelerated (and their effects on the SDGs targets). To do so, we use an interdisciplinary approach – the modelling is an important component and incorporates the policies and targets, but we also combine this with socio-technical analysis which focuses more on the impediments and enabling conditions for transformation. The crisis framing is important in terms of these enabling conditions, as it is a well-known precursor and enabler for transformation – providing “windows of opportunity”

for governments to implement policy reforms. The COVID crisis is also incorporated as a 'shock' into the modelling and features in the storylines. However, we have removed reference to crisis from the abstract as we agree that it is not the key finding from our study.

There are a few chickens and a few eggs. First, there are pathways that have been identified that allow us to understand how to pursue the SDGs. That comes first. So, is this manuscript really identifying *how* to pursue these pathways? I don't get that message from the manuscript. Is the manuscript telling me that a crisis space has facilitated the achievement of these results? That's not really covered in the manuscript either, and certainly isn't the main point.

Response: thank you for this comment and the reviewer is correct that the crisis space is not the key finding we wish to highlight, but rather is a part of the modelling and storylines. We have now revised the abstract and discussion to better reflect the key findings which we wish to highlight and which also align with the study aims mentioned above – i.e. policy packages for accelerating each transformation, their effects on achieving the SDGs, synergies/trade-offs, and impediments and conditions that determine policy success. We refer the reviewer to the revised abstract and revised discussion section (from line 479).

If the point is to say that the SDGs can be achieved if you pursue these practical policies, then you need to focus on the specific interventions that get you to the scenarios you're building. How do we know that these policies are easily achieved? What is the magnitude of the interventions we're pursuing and how do we know that's a reasonable target? The paper is now about target setting and policies in relation to Australia's pursuit of these global pathways.

Response: thank you for this comment. In response to this and other comments, we have revised the abstract and introduction to improve clarity regarding the aims of the study. This includes advancing knowledge on the policy packages that accelerate each transformation and their effects on the SDGs. These policies are introduced into the modelling for the STP – as mentioned from line 205 onwards – and cover a broad mix of policies including tax reform, additional spending on social services and transfers, and scaled up investment in sustainable energy, industry, transport, agriculture, and infrastructure. As noted in the manuscript, this requires additional annual government expenditure of 7.5% on average which is financed through additional tax/revenue measures (as depicted in Supplementary Figure 1(d)). Given that we are modelling six transformations and all 17 goals, the scope of policy settings is considerable. To ensure transparency and provide the necessary detail to those readers who are interested in exploring the specific policy settings in more depth, we include all of the policies in Supplementary Table 3. We have also now revised the discussion section so that important policy findings are clearer.

Is this a paper about synergies and trade-offs? If so, that should be the focus. You talk about the energy system as being essential and highly synergistic. That's a potentially useful finding, but then the abstract is focused on that.

Response: thank you for this comment. As noted, we have revised the abstract and introduction to improve clarity regarding the aims of the study. We do indeed seek to advance knowledge on important synergies and trade-offs between the six transformations and the SDGs, and we believe the study provides novel insights in this regard. The reviewer is correct that energy decarbonization is particularly synergistic. We now also reflect these findings in the revised abstract.

I think R3 also has good points about the framing being more focused on a country level application of a global development pathway framework. There is less in this piece than I would expect about how a global framework could be applied to a national framework. And this global framework is grey

literature (I believe), so that is an additional challenge. It's an especially important omission because the SDGs in high income countries are extremely different than LDCs both in narrative and empirics.

Response: thank you for this comment. In response, we have now revised the introduction and the discussion to ensure that the national aspect is clearer in the study framing. It is true that the six transformation entry points are published in a global report – the UN Global Sustainable Development Report (GSDR). Our study includes authors of the 2023 GSDR which uses the entry points framework. However, it is also important to clarify here that the entry points framework is intended to be used by countries to support their national implementation – as are the SDGs themselves. While priorities and circumstances vary across countries, the intent of the six entry points is to help governments to simplify the very broad and complex scope of the SDGs into six main areas for action.

The GSDR is grey literature, but it is also peer reviewed – the 2023 GSDR was peer reviewed by >100 academics coordinated through the International Science Council. It is also a major United Nations science-policy publication which is intended to assist countries to implement the SDGs. As such, it is robust, relevant and has considerable merit, particularly for use in national research on how to accelerate progress on the SDGs. A very similar framework of 'six transformations' has also been published in the academic literature (i.e. Nature Sustainability (Sachs et al, 2019)), again with a focus on national implementation.

I think there is a tremendous amount of "meat" to this analysis and it should be published. But you need to clarify the purpose of the analysis earlier, more clearly, and make broader changes to reflect these concerns.

Response: thank you for this feedback. As noted above, we have now clarified the purpose of the analysis earlier and more clearly – in the abstract and introduction – and shaped our key findings in the discussion around the study aims. We hope that these revisions adequately address the comments and concerns raised.

Minor comments relate to redundancies in the text (R3 is right about this, I think) and Figure 3: the line graphs show a spike or dip in the year the scenarios take that doesn't make sense. Why does the economy fall in the BBS scenario so quickly? Why the spike in food availability?

Response: we have reviewed the text again to identify redundancies and have made several revisions in track changes to the manuscript to remove repetition. With regard to the figures, the spikes are associated with COVID impacts and the large COVID stimulus response. As we note in the manuscript (paragraph starting from line 168 onwards) we incorporate the shock of the COVID lockdowns into the model as well as the government's large COVID stimulus package of around 15% of GDP, much of which was spent on social welfare transfers to households. This had a large but temporary impact on indicators such as poverty rates and inequality (in T2) as well as population below the food poverty line and nutrition-related indicators (in T3). Under the BBS, this stimulus is discontinued so the effect is temporary as shown by the spike. For STP, some of the stimulus is continued through to 2030.

REVIEWERS' COMMENTS

Reviewer #4 (Remarks to the Author):

I believe the authors have sufficiently responded to the feedback and that this manuscript should be published.

REVIEWERS' COMMENTS

Reviewer #4 (Remarks to the Author):

I believe the authors have sufficiently responded to the feedback and that this manuscript should be published.

Response: thank you for taking the time to review our manuscript and providing helpful feedback. No further revisions required.